# Feld-induced modulation of two-dimensional electron gas at LaAlO₃/SrTiO₃ interface by polar distortion of LaAlO₃

Jinsol Seo[1], Hyungwoo Lee[2,3], Kitae Eom [2], Jinho Byun[1], Taewon Min[4], Jaekwang Lee [4], Kyoungjun Lee[2], Chang-Beom Eom[2] & Sang Ho Oh [1] ✉

Since the discovery of two-dimensional electron gas at the LaAlO₃/SrTiO₃ interface, its intriguing physical properties have garnered significant interests for device applications. Yet, understanding its response to electrical stimuli remains incomplete. Our in-situ transmission electron microscopy analysis of a LaAlO₃/SrTiO₃ two-dimensional electron gas device under electrical bias reveals key insights. Inline electron holography visualized the field-induced modulation of two-dimensional electron gas at the interface, while electron energy loss spectroscopy showed negligible electromigration of oxygen vacancies. Instead, atom-resolved imaging indicated that electric fields trigger polar distortion in the LaAlO₃ layer, affecting two-dimensional electron gas modulation. This study refutes the previously hypothesized role of oxygen vacancies, underscoring the lattice flexibility of LaAlO₃ and its varied polar distortions under electric fields as central to two-dimensional electron gas dynamics. These findings open pathways for advanced oxide nanoelectronics, exploiting the interplay of polar and nonpolar distortions in LaAlO₃.

The concept of confining electrons dimensionally or geometrically, and modulating their density with an electric field, is pivotal in many emerging devices. For two-dimensional electron confinement, potential wells created at the junction of two dissimilar materials are extensively utilized[1,2]. Traditional potential wells, resulting from energy band alignment, typically extend over several tens of nanometers. However, the LaAlO₃/SrTiO₃ polar/nonpolar oxide interfaces demonstrate electron confinement within a few unit cells (u.c.), but with a higher density, heralding new possibilities in oxide nanoelectronics[3–6]. This phenomenon, known as the two-dimensional electron gas (2DEG), arises to cancel the polar field generated by the polar discontinuity at the interface[7]. The 2DEG, confined to a few u.c. beneath the interface, is intrinsically linked to the atomic structure, resulting in a variety of emergent physical properties[8–10]. This intimate connection between the atomic structure and electronic properties at

the LaAlO₃/SrTiO₃ interface exemplifies the potential for novel electronic behaviors in nanostructured materials.

It has been shown that applying an electric field perpendicular to the LaAlO₃/SrTiO₃ heterostructure interface can modulate its conductivity. This field-induced modulation of the 2DEG has led to the successful fabrication of various prototype devices, including field effect transistors[11–13] and memristors[14–16]. However, the underlying mechanisms behind this modulation remain unclear. Two major models have been proposed based on: the electrostatic response of the potential well[11] and the electromigration of oxygen vacancies (V_O)[17,18], but experimental validation is lacking. V_O is known to form at the surface of the LaAlO₃/SrTiO₃ heterostructure, contributing to 2DEG formation[19]. Under high electric fields, V_O may migrate from the LaAlO₃ surface toward the LaAlO₃/SrTiO₃ interface, influencing the density and/or distribution of 2DEG[20,21]. This migration has been

[1]Department of Energy Engineering, KENTECH Institute for Energy Materials and Devices, Korea Institute of Energy Technology (KENTECH), Naju, Republic of Korea. [2]Department of Materials Science and Engineering, University of Wisconsin-Madison, Madison, Wisconsin, USA. [3]Department of Energy Systems Research and Department of Physics, Ajou University, Suwon, Republic of Korea. [4]Department of Physics, Pusan National University, Busan, Republic of Korea. ✉e-mail: shoh@kentech.ac.kr

particularly linked to the resistive switching or memristor-like behaviors seen in devices with hysteresis in their current-voltage (I-V) characteristics[16]. Conversely, devices that do not exhibit resistive switching[11,22] may undergo 2DEG modulation via a different mechanism. Understanding these underlying processes is crucial for advancing oxide nanoelectronics, as they determine device behavior and efficiency.

We propose that the polar distortion of LaAlO₃, induced by an electric field, should be recognized as a key factor in the modulation of the 2DEG at LaAlO₃/SrTiO₃ inteface. It has been shown that various polar phases of LaAlO₃ can be stabilized on SrTiO₃ substrate with varying extents of anti-ferrodistortive rotation and polar distortion of $AlO_6$ octahedron depending on the film thickness, interface orientation, and the internal electric field[19,23]. Especially, when the electric field is applied to the LaAlO₃ film, the polar distortion tends to evolve preferentially as a consequence of the depolarization effect[24,25]. One example is the evolution of polar distortion in a subcritical LaAlO₃ film, which arises to alleviate the internal polar field in the LaAlO₃/SrTiO₃ heterostructure in the absence of 2DEG[19,26,27].

The characterization of 2DEGs in oxide systems requires a charge-sensitive imaging technique with high spatial resolution. The phase contrast techniques based on transmission electron microscopy (TEM) that measure the phase shift of the electron beam as it passes through the confined electrons have proven their capability in the imaging of 2DEG[28–30]. The imaging of the field-induced modulation of 2DEG requires the application of this technique in situ under electric fields. Furthermore, to resolve the mechanism of 2DEG modulation multiple types of information that can address the spatial distribution of $V_O$ and the evolution of polar distortion should be acquired simultaneously with the modulation of 2DEG under an electric field.

Here we show the comprehensive in-situ analysis of a LaAlO₃/SrTiO₃ 2DEG device under electrical bias in TEM. The in-situ inline electron holography successfully visualized the field-induced modulation of 2DEG at the LaAlO₃/SrTiO₃ interface[31]. While in-situ electron energy loss spectroscopy (EELS) confirmed no measurable electromigration of $V_O$, atom-resolved scanning transmission electron microscopy (STEM) imaging revealed that the polar distortion evolves in the LaAlO₃ film and exhibits field-induced switching behavior. The field-induced polar distortion adds additional polarization charges at the LaAlO₃/SrTiO₃ interface, governing the field-induced modulation of 2DEG. Our in-situ TEM study, as opposed to the previously suggested mechanism based on $V_O$[17,18], demonstrates that the flexibility of LaAlO₃ that exhibits various polar distortions under electric fields is key to the modulation of 2DEG at the LaAlO₃/SrTiO₃ interface.

## Results
### Device structure and characteristics
A model device structure based on LaAlO₃/SrTiO₃ heterostructure that allows the modulation of 2DEG by the applied field was prepared on Nb-doped SrTiO₃ (001) substrate by epitaxial growth of the following layers; 3 u.c. LaAlO₃ layer for charge blocking, 10 u.c. LaAlO₃/15 u.c. SrTiO₃ (001) system for 2DEG formation, and conductive SrRuO₃ for top electrode (Fig. 1a). We confirmed that the LaAlO₃/SrTiO₃ interface has the $n$-type LaO/TiO₂ termination by atomic-resolution STEM energy dispersive X-ray spectroscopy (Supplementary Fig. 1), a prerequisite for the formation of 2DEG[7]. The 3 u.c.-LaAlO₃ layer was inserted between 15 u.c.-SrTiO₃ and conductive Nb: SrTiO₃ substrate to suppress the overflow of electrons from the conductive Nb:SrTiO₃ to the insulating SrTiO₃. In this oxide heterostructure, except for the conductive SrRuO₃ top electrode and the Nb:SrTiO₃ substrate, there are two major sources of local charges at the interfaces, which are 2DEG at the LaAlO₃/SrTiO₃ and $V_O$ at the LaAlO₃ surface in contact with SrRuO₃ electrode. We confirmed that the latter acts as the source of 2DEG[19].

Considering that the formation of 2DEG at LaAlO₃/SrTiO₃ interface is contingent upon the crystalline quality of SrTiO₃, the application of

SrTiO₃ film might pose limitations on the formation of 2DEG[32–35]. To investigate the electrical transport of the 10 u.c. LAO/15 u.c. SrTiO₃ (001) interface used in this study, we fabricated a similar structure on an undoped SrTiO₃ substrate, but without the SrRuO₃ top electrode. The transport characterization, including interface charge density, resistance, and mobility measurements, confirmed the formation of 2DEG at the LaAlO₃/SrTiO₃ interface, as shown in Supplementary Fig. S2. When compared to a standard LaAlO₃ (10 uc)/SrTiO₃ substrate sample (Supplementary Fig. S3), the 2DEG density in our model device structure was found to be lower, and its resistance was higher. This disparity is attributable to unavoidable crystal imperfections in the SrTiO₃ film.

The TEM samples for the in-situ electrical biasing were prepared by using a focused ion beam (FIB) (Supplementary Fig. S4). To induce an electric field perpendicular to the LaAlO₃/SrTiO₃ interface, the voltage was applied to the SrRuO₃ top electrode with the Nb:SrTiO₃ substrate being held at the electrical ground. The potential distribution from the SrRuO₃ top electrode to the Nb:SrTiO₃ substrate for each applied voltage was confirmed by measuring the phase shift of the electron beam using off-axis electron holography (Supplementary Fig. S5)[30,36]. After testing multiple TEM samples one that exhibits the lowest current up to ± 3 V was chosen for further detailed in-situ STEM characterization (Supplementary Fig. S6).

### Visualization of field-induced charge modulation
To visualize the 2DEG and trace its field-induced modulation, the phase shift of the transmitted beam, which varies sensitively with the electrostatic potential, was measured precisely with a high signal-to-noise ratio. We used inline electron holography to reconstruct the phase shift of the transmitted beam from a focal series of bright-field TEM images[31]. In this method, and other phase contrast electron microscopy methods as well[29,37], the transmitted beam undergoes the inherent phase shift as it passes through a sample (even though charge-neutral and non-magnetic) due to the positive background potential of crystal which is known as the mean inner potential ($V_0$) (Supplementary Fig. S7)[38–40]. Due to the different $V_0$, the measured potential of each layer in the heterostructure differs by a few eV even before the electrical biasing. For example, the LaAlO₃ layers are distinguished from SrRuO₃ and SrTiO₃ layers by ~2–3 V larger $V_0$ as this material consists of a heavy element, La. Moreover, as the potential variation across the interfaces is dominated by the difference of $V_0$ ($\Delta V_0$), it is not straightforward to discern one that originates from local charges ($V_Q$) confined to the interfaces. We have developed a model-based interpretation that takes account of both $V_0$ and $V_Q$ (Supplementary Fig. S8)[41]. The modeled potential profile shown in Supplementary Fig. S8e agrees well with the measured one (refer to Fig. 1c).

The in-situ TEM biasing was performed by applying a DC voltage to the SrRuO₃ electrode, ranging from − 3 V to + 3 V in 1 V increments at room temperature, with the conductive Nb:SrTiO₃ substrate grounded. The potential maps obtained at − 3 V, 0 V, and + 3 V demonstrate a clear potential difference in the SrRuO₃ top electrode, which is not observed in the Nb:SrTiO₃ (Fig. 1b). The internal potential within the heterostructure sandwiched by these electrodes is modified locally by the applied voltage. Using the potential in the unbiased state as a reference, we monitored deviations induced by the applied voltage from this reference potential[42]. For example, in the 15 u.c.-SrTiO₃ layer where 2DEG exists, the potential becomes more negative under +3 V (color change to blue) but increases under −3 V (to green) from the unbiased state. The specifics of how the potential varies with the applied voltages become clearer in the potential profiles plotted across the heterostructure (Fig. 1c). The most notable change is evident in the local curvature of the potential profiles within the SrTiO₃ region (as indicated by the radius of the circles in Fig. 1c), which directly correlates with the modulation of local charges in that area.

The potential map can be directly converted to the charge density map by applying a 2D Laplacian image filter, which is simply

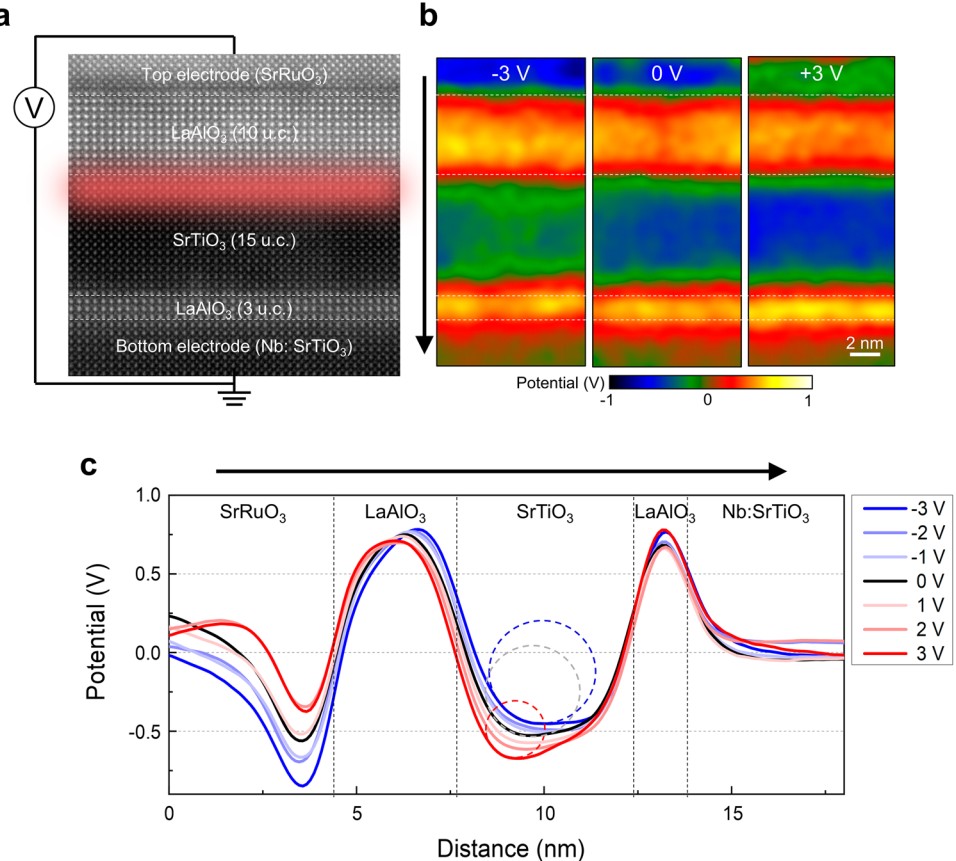

**Fig. 1 | Potential distribution in LaAlO$_3$/SrTiO$_3$ heterostructure device measured by inline electron holography under applied voltages. a** HAADF STEM image showing the device structure. The LaAlO$_3$ (10 u.c.)/SrTiO$_3$ (15 u.c.) interface where 2DEG forms is highlighted in red. Note that the LaAlO$_3$/SrTiO$_3$ interface satisfies the requirements for 2DEG formation; the LaO/TiO$_2$ interface termination is *n*-type, and the LaAlO$_3$ (10 u.c.) is thicker than the critical thickness (4 u.c.). The LaAlO$_3$/SrTiO$_3$ heterostructure was grown on top of the LaAlO$_3$-buffered Nb:SrTiO$_3$ (001) substrate. The SrRuO$_3$ and the Nb:SrTiO$_3$ substrate serve as the top and bottom electrodes, respectively. To apply an electric field perpendicular to the LaAlO$_3$/SrTiO$_3$ interface, DC voltage was applied to the SrRuO$_3$ top electrode while

the Nb:SrTiO$_3$ is electrically grounded. **b** Potential maps obtained by inline electron holography under the applied voltage of −3 V (left) and +3 V (right) to the SrRuO$_3$ top electrode. Among the internal layers sandwiched by the electrodes, the 15 u.c.-SrTiO$_3$ layer where 2DEG exists shows the most pronounced change in the potential distribution. **c** Potential profiles drawn across the heterostructure along the arrow in (**b**). Potential variation is pronounced most at the SrTiO$_3$ side of the LaAlO$_3$/SrTiO$_3$ interface as indicated by the dash circles representing the curvature of the potential profile, which is directly related to the charge density according to the Poisson's equation.

the mathematical implementation of Poisson's equation (note that for more rigorous treatment the dielectric constant of each layer should be calibrated by considering their field-dependency). This image processing, when applied to the phase image of inline electron holography, is capable of producing a charge density map with a high signal-to-noise ratio (Fig. 2a) owing to the good transmittance of high spatial frequency information of phase[41]. In the resulting charge density maps and profiles all heterointerfaces where $V_0$ changes abruptly display a pair of peaks with opposite signs, irrespective of the presence of localized charges (Fig. 2a, b). These signals originating from $V_0$-difference ($\Delta V_0$) across the interfaces are unavoidably convoluted with the real interface charge signals, obscuring the intuitive analysis. Nonetheless, some of the interface signals do respond to the applied voltage and exhibit noticeable changes, as evidenced by the change of local curvature in the LaAlO$_3$/SrTiO$_3$ heterostructure region in the potential profiles (Fig. 1c). Assuming that the $\Delta V_0$ remains unchanged under bias, a change in the interface signal is primarily due to the modulation of charges confined at the interface.

To investigate the interface charge modulation, the charge density profiles obtained at various voltages were overlaid onto that of an unbiased state after the careful alignment (Fig. 2b). The charge density profile is altered noticeably at the LaAlO$_3$/SrTiO$_3$ interface with 2DEG

but not at the rest of the interfaces in the heterostructure. A close-up of the charge density maps (Fig. 2c) and profiles across the LaAlO$_3$/SrTiO$_3$ interface (Fig. 2d) reveals a characteristic change with the applied voltage; under positive voltage, the peak in the SrTiO$_3$ side of the interface where the 2DEG exists shifts gradually toward the LaAlO$_3$ with an increase of the peak area; under negative voltage, the peak area is reduced while it moves away from the interface.

To elucidate how the observed changes are related to the modulation of interface charges, we carried out the modeling of charge density profiles by varying the density of interface charges (Supplementary Fig. 9). The change in the density of 2DEG results in the change of the peak area of the SrTiO$_3$ side of the interface signals, i.e., the higher the 2DEG density, the larger the peak area. As such, the interface signal on the SrTiO$_3$ side can be used as a qualitative descriptor of 2DEG modulation. The observed change of the charge density profile under positive bias can be interpreted such that the 2DEG gets attracted to the LaAlO$_3$/SrTiO$_3$ interface, increasing the 2DEG density in proportion with the applied voltage. Under negative bias, the distribution of 2DEG gets broadened with the decrease in the density. However, note that due to the finite spatial resolution of the technique (~0.7 nm) the change in the 2DEG density in the SrTiO$_3$ results in the change of the peaks across the LaAlO$_3$/SrTiO$_3$ interface (Fig. 2f and Supplementary Fig. S9f). For the quantitative measurement

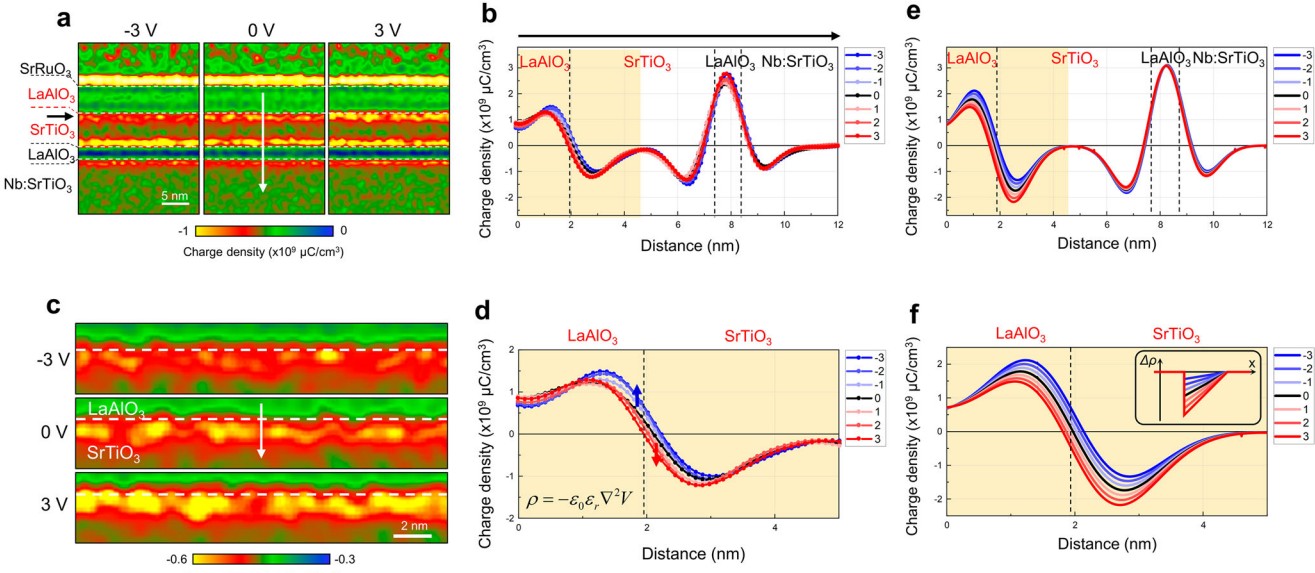

**Fig. 2 | Charge density maps and profiles of LaAlO₃/SrTiO₃ heterostructure device obtained by inline electron holography under applied voltages.**
**a** Charge density maps derived from the inline electron holography potential maps obtained under the applied voltage of −3 V (left), 0 V (middle), and +3 V (right). The position where 2DEG exists is marked by the black arrow. **b** Charge density profiles obtained across the heterostructure (along the white arrow in (**a**)) under various applied voltages. All heterointerfaces across which $V_O$ changes abruptly exhibit a pair of peaks with opposite signs. The LaAlO₃/SrTiO₃ interface where 2DEG exists shows noticeable changes in the profiles with applied voltages. **c** Magnified view of the charge density maps, and **d** profiles of the LaAlO₃/SrTiO₃ interface region (highlighted in yellow in (**b**)). Arrows in (**d**) indicate the profile change across the LaAlO₃/SrTiO₃ interface under positive (red) and negative (blue) voltages. **e** Charge

density profiles obtained from the charge model given in Supplementary Fig. S8. The model takes account of the interface charges ($V_O$ at the LaAO₃ surface and the 2DEG at the LaAlO₃/SrTiO₃ interface) that compensate for the polar field in the LaAO₃ layer and the $V_O$ in each layer. **f** Magnified view of the charge density profiles of the LaAlO₃/SrTiO₃ interface region (highlighted in yellow in (**e**)). Variation of the 2DEG density in the SrTiO₃ side of the interface (inset) was considered as a response to the applied voltages. Due to the finite resolution of the inline electron holography technique, the change of the charge density profiles due to the 2DEG modulation is extended across the LaAlO₃/SrTiO₃ interface to the LaAlO₃, in good agreement with the experimental results. Details of the modeling are given in Supplementary Figs. S8 and S9.

of charge modulation, therefore, the areal integration was conducted over an extended distance across the interface and the corresponding value at unbiased state was subtracted. The total amount of additional charge density increased (decreased) at + 3 V (− 3 V) from the unbiased state is measured to be about 0.41 $e/a^2$, where $a$ is the lattice parameter. We note that the charge density measured by inline electron holography is the density of net charges encompassing all charges, not only the 2DEG which contributes to the transport but also other localized or trapped charges that do not contribute to the transport. Considering that some extent of cation intermixing between LaAlO₃ and SrTiO₃ is unavoidable at the interface[19], it is likely that some additional electrons trapped near the interface region of LaAlO₃ are also included in the inline holography data.

### Field-induced modulation of 2DEG

The field-induced modulation of the 2DEG can be assessed by analyzing changes in the fine structure of the EELS Ti-L₂,₃ edge in response to varying applied voltages[43] (Fig. 3 and Supplementary Fig. S10). It is well-established that the 2DEG predominantly occupies empty Ti 3$d$ orbitals, thereby reducing the valence state of Ti from 4 + to 3 +. Specifically, at the LaAlO₃/SrTiO₃ (001) interface, the 2DEG preferentially fills the $d_{xy}$ orbital within the $t_{2g}$ band, the lowest energy state[42]. Applying an electric field can alter the 2DEG density, either increasing to or decreasing the electrons localized in the Ti 3$d$ states. These changes in the 2DEG density are reflected in the variation of the Ti³⁺ fraction with applied voltages, particularly observable through changes in the intensity of the $e_g$ peaks at both Ti-L₃ and L₂ edges, as these electron states are sensitive to changes detected by EELS (Supplementary Fig. S10). Since the features of both Ti³⁺ and Ti⁴⁺ state co-exists in the EELS Ti-L₂,₃ edge to varying extents, the Ti³⁺ fraction has been quantified using multiple linear least squares (MLLS) fitting of the EELS Ti-L₂,₃ edge with reference

spectra for Ti³⁺ and Ti⁴⁺ (Supplementary Fig. S11). The fraction of Ti³⁺ decreases gradually from the maximum value at the interface and is hardly observable beyond the 5ᵗʰ layer. The spatial profiles of the measured Ti³⁺ fraction for each applied voltage are presented in Fig. 3d. The change of the Ti³⁺ profiles with the applied voltage agrees qualitatively well with the holography results in that the 2DEG concentration increases (decreases) under positive (negative) bias.

We note that the incremental charges detected by the change in the Ti valence state (Ti³⁺) from EELS Ti-L₂,₃ differ from those measured by inline electron holography, as these two techniques probe fundamentally different types of charges. The inline electron holography measures the net charge density, encompassing all charges, whereas the charges detected by the change of Ti valence state to Ti³⁺ from EELS Ti-L₂,₃ edge specifically reflect electrons localized within the Ti −3$d$ orbitals. Consequently, the incremental 2DEG density assessed by the change of Ti valence state from EELS Ti-L₂,₃ edge is typically smaller than that measured by inline electron holography. The latter captures a broader spectrum of net charges distributed across the somewhat blurred LaAlO₃/SrTiO₃ interface, including some additional charges trapped near the interface region of LaAlO₃.

### Oxygen vacancies under electric fields

As to the origin of the field-induced modulation of 2DEG, we first considered the electromigration of $V_O$ in the heterostructure. To detect $V_O$, the EELS O-K edge was used as it varies sensitively with $V_O$[19]. Through the fine structure analysis of the EELS O-K edge, we confirmed the presence of $V_O$ in the LaAlO₃ surface region (Supplementary Fig. S10b and c), which acts as the source for 2DEG, in accord with the previous studies[19,23] However, the in-situ characterization of the EELS O-K edge fine structure under voltages up to +/− 3 V shows that the spatial distribution of $V_O$ remains almost the same, indicating that the

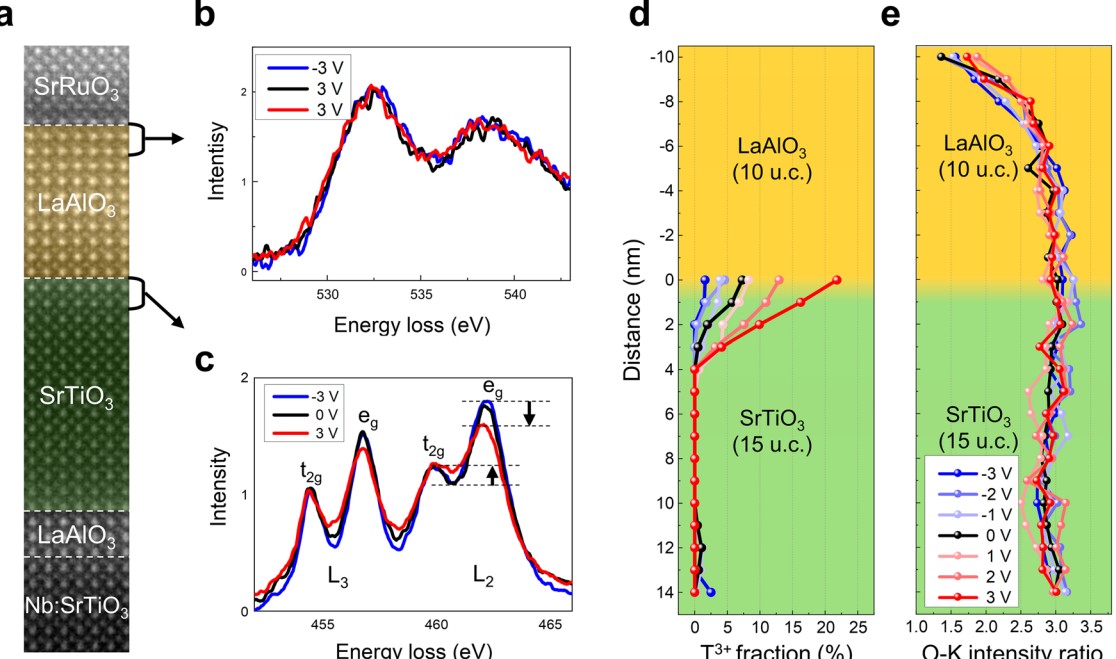

**Fig. 3 | In-situ STEM-EELS analysis on the interface charges in LaAlO₃/SrTiO₃ heterostructure under applied voltages. a** HAADF STEM image of the heterostructure. **b** EELS O-K edge obtained from the near-surface region of the LaAlO₃ under − 3 V (blue), 0 V (black), + 3 V (red). The analysis of the fine structure (Supplementary Fig. S10b, c) indicates the presence of $V_O$ at the LaAlO₃. The fine structure of the O-K edge EEL spectra remains almost identical without noticeable change under the applied voltages. **c** EELS Ti-$L_{2,3}$ edge obtained from the SrTiO₃ side of interface under − 3 V (blue), 0 V (black), + 3 V (red). The fine structure of EELS Ti-$L_{2,3}$ edge changes with the applied voltages, e.g., under + 3 V the relative intensity of the $e_g$ peak decreases, and the valley between $t_{2g}$ and $e_g$ peaks increases (black arrows). Note that the Ti-$L_{2,3}$ edge within the SrTiO₃ layer far from the interface does not change under the applied voltages (Supplementary Fig. S10d). **d** Plot of Ti³⁺ fraction determined by MLLS fitting of EELS Ti-$L_{2,3}$ edge using the reference spectra of Ti³⁺ and Ti⁴⁺ (Supplementary Fig. S11). The fraction of Ti³⁺ near the interface increases (decreases) under positive (negative) voltages, which is consistent with the charge modulation measured by inline holography. **e** O-K intensity ratio plotted across the LaAlO₃/SrTiO₃ interface. The integrated intensity of the O-K edge was normalized by that of the La-$M_{4,5}$ and Sr-$L_{2,3}$ edge for LaAlO₃ and SrTiO₃, respectively. The oxygen deficiency due to $V_O$ near the LaAlO₃ surface remains unchanged up to +/− 3 V.

electromigration of $V_O$ is not activated (Fig. 3b). The integrated intensity of EELS O-K edge normalized by the A-site ions (Sr and La) also shows a similar trend; the distribution of $V_O$ near the surface of LaAlO₃ remains unaltered without electromigration (Fig. 3e). The in-situ EELS characterization clearly shows that while the 2DEG modulates in response to the applied field, the $V_O$ does not undergo electromigration, demonstrating that the electromigration of $V_O$ is not attributable to the field-induced modulation of 2DEG.

The formation of $V_O$ and the associated 2DEG is a response to cancel the polar field, which arises due to the polar discontinuity at both the LaAlO₃ surface and the LaAlO₃/SrTiO₃ interface. Consequently, these charges ought to be spatially confined to their respective boundaries as fixed charges. If not, the previously canceled polar field would become disturbed. Considering the applied electric field is in the order of $10^6$ V/cm, perhaps a larger electric field may drive the electromigration of $V_O$.

The spatially confined $V_O$ at the LaAlO₃ surface can influence the band alignment and contact at the interface with the electrode (here SrRuO₃). The work function of SrRuO₃ and LaAlO₃ is 5.2 eV and 4.8 eV, respectively[14,44]. This work function difference (0.4 eV) is expected to introduce a Schottky barrier at the interface. The charged $V_O$ can result in the pinning of the Fermi level and/or lowering of the barrier height[45,46], affecting the transport across the interface. The analysis of the I-V curves measured during in-situ STEM biasing experiments (Supplementary Fig. S12) provides a clue to the conduction mechanism and thereby the characteristic of the SrRuO₃/LaAlO₃ contact. Choosing the most reliable I-V curve from multiple TEM samples which can represent the device characteristics (the black curve in Supplementary Fig. S6b, Fig. S12a), we analyzed the curve according to the various transport mechanisms. Among various approaches to fitting this I-V

curve, the best match was obtained by the log I-log V plot. This plot clearly demonstrates a space-charge-limited conduction behavior, which is governed by trap sites. Based on these I-V characteristics, the SrRuO₃/LaAlO₃ interface can be characterized as an Ohmic contact, with the trap sites primarily originating from $V_O$.

## Polar distortion of LaAlO₃ under electric field

We found that the field-induced ionic polarization ($P_i$) of the LaAlO₃ layer is the major source of the modulation of 2DEG. The $P_i$ was determined on the u.c. basis by measuring the displacement of the B-site ions ($\delta_B$) from the center of the A-site sublattice in the HAADF STEM images (Fig. 4a). The contrast of oxygen columns was too weak to measure their displacements ($\delta_O$) directly from the HAADF STEM images. As the displacement of oxygen is not ignorable and thus must be included in the calculation of $P_i$, we have gone through the literature for reasonable treatment of oxygen displacement[47–51]. For the given displacement of the B-site cation, it was assumed that the displacement of oxygen is twice as large in the opposite direction. As to the charges of oxygen and cations for the calculation of polarization, we used the effective charges reported in the literature[52].

Before applying the bias the $P_i$ measured in both the LaAlO₃ and SrTiO₃ layers were marginal (Fig. 4b). It is well known that when the LaAlO₃ film grows thicker than the $t_c$ for 2DEG formation, like the 10 u.c. film in this study, the initial polar distortion is mitigated, giving way to an antiferrodistortive rotation[19,23,26]. However, when an external field is applied, this scenario may not hold. Under the applied field the polar distortion, and accordingly $P_i$, evolves in the LaAlO₃ as a consequence of the depolarization effect that counterbalances the applied field[24,25]. This occurs because the interface charges, namely the 2DEG and $V_O$, are bound charges that cannot migrate or redistribute freely

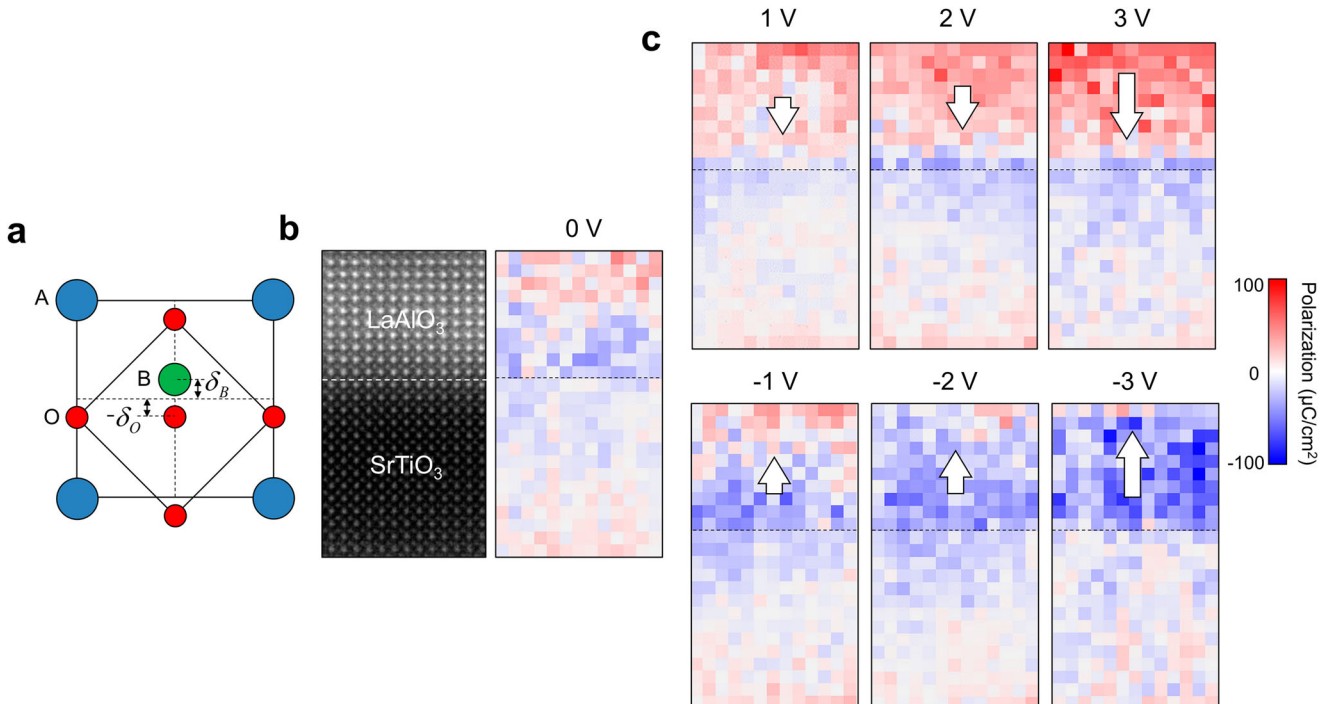

**Fig. 4 | Ionic polarization induced in LaAlO₃/SrTiO₃ heterostructure by electric field. a** Atomic model of ABO₃ perovskite oxide depicting the ionic displacements. The displacement of B-site cation and oxygen from the center of the A-site cation sublattice is defined as δ_B and δ_O, respectively. **b** Polarization (P_i) map of the LaAlO₃/SrTiO₃ heterostructure determined using HAADF STEM recorded at unbiased condition (0 V). δ_B was measured directly from the HAADF STEM image but δ_O was approximated as two times the δ_B in the opposite direction. For the calculation of polarization, the effective charges reported in the literature are used[52]. Only the out-of-plane component of polarization along the field direction was displayed in the map. **c** P_i maps obtained under applied voltage conditions from +/− 1 V to +/− 3 V. The P_i is induced in the LaAlO₃ layer along the direction of the applied field. The P_i signal appearing in the SrTiO₃ is the buckling associated with the accommodation of 2DEG[19,53,54].

under the influence of an applied field. In the LaAlO₃, $P_i$ increases proportionally with the applied voltages (Figs. 4c, 5a), and notably, the direction of $P_i$ undergoes 180° switching when the electric field is reversed, exhibiting behavior akin to a ferroelectric, as illustrated in Fig. 4c (Supplementary Fig. S13).

We note that the SrTiO₃ layers beneath the LaAlO₃/SrTiO₃ interface exhibits a noticeable polar distortion (blue pixels in Figs. 4b, c and 5a), which is related to the accommodation of 2DEG[19,53,54]. The electrons injected from the V_O at the LaAlO₃ surface to the interface are confined to the near-interface SrTiO₃ region by band bending. The gradient of band-bending in this region results in the electric field which causes the polar distortion of SrTiO₃ layers (Supplementary Fig. S14) as also seen in our previous work[19].

As to the octahedral tilts in the LaAlO₃/SrTiO₃ (001) system, we cannot determine the octahedral tilts experimentally in the LaAlO₃ and SrTiO₃ layers using HAADF STEM images due to the weak oxygen contrast. It has been known that the octahedral tilts and polar distortion are not compatible but compete with each other in the LaAlO₃/SrTiO₃ (001) system[23,26]. When the polar distortion becomes dominant in the LaAlO₃ layers due to the applied field as in the present study or by the internal polar field below the $t_c$ for 2DEG formation[27], the octahedral tilts may not coexist with the polar distortion but tend to disappear. Indeed, we could not observe the octahedral tilts except the polar distortion in the SrTiO₃ layers in the DFT calculation.

The $P_i$ arising in the LaAlO₃ layer under the electric field adds extra charges to the LaAlO₃/SrTiO₃ interface in addition to the ordinary dielectric charges. These charges bound to the LaAlO₃/SrTiO₃ interface are the major source that modulates 2DEG. For example, the downward polarization pointing to the interface ($P_i^-$) under positive voltage results in the positive bound charge at the interface, which attracts the 2DEG and increases its density. The upward polarization ($P_i^+$) deposits

the negative bound charge at the interface, which repulses the 2DEG and decreases the density. The magnitude of $P_i$ averaged over the LaAlO₃ layer is 23.8 μC/cm² and − 32.4 μC/cm² at + 3 V and − 3 V, respectively (Fig. 5b).

The net charges induced in the LaAlO₃, i.e., the dielectric charges ($\varepsilon_0\varepsilon_r E_a$) and the polarization charge ($P_i$), are comparable to the total amount of net charge modulation ($\Delta\sigma$) measured by the inline electron holography (Fig. 5c), therefore setting up a charge balance equation as follows[55]:

$$\Delta\sigma = \varepsilon_0\varepsilon_r E_a + P_i. \tag{1}$$

Once the electric field is removed, the $P_i$ in the LaAlO₃ layer relaxes and disappears without leaving a measurable remnant polarization, demonstrating the flexibility in varying the polar state of LaAlO₃ with the applied field in the LaAlO₃/SrTiO₃ heterostructure. To fully exploit the large but volatile $P_i$, and its interaction with 2DEG at the interface, charged defects such as those originating from interface roughness and cation intermixing, should be suppressed.

The electrons transporting between the two electrodes (SrRuO₃ and Nb:SrTiO₃) are the major source leading to the modulation of the 2DEG at the LaAlO₃/SrTiO₃ interface. These charges are attracted to or repulsed from the LaAlO₃/SrTiO₃ interface to compensate the bound charges arising in the LaAlO₃ from both ordinary dielectric response and additional ionic polarization of LaAlO₃. The former, represented by a blue bar in Fig. 5c, is calculated by using $\varepsilon_0\varepsilon_r E_a$. The latter corresponds to the charges from the measured $P_i$ in LaAlO₃. The summation of the two contributions ($\varepsilon_0\varepsilon_r E_a + P_i$), represented by the red bar in Fig. 5c, constitutes the net bound charges. The electrons transporting between the two electrodes are attracted to or repulsed from the LaAlO₃/SrTiO₃ interface to compensate for the net bound charges.

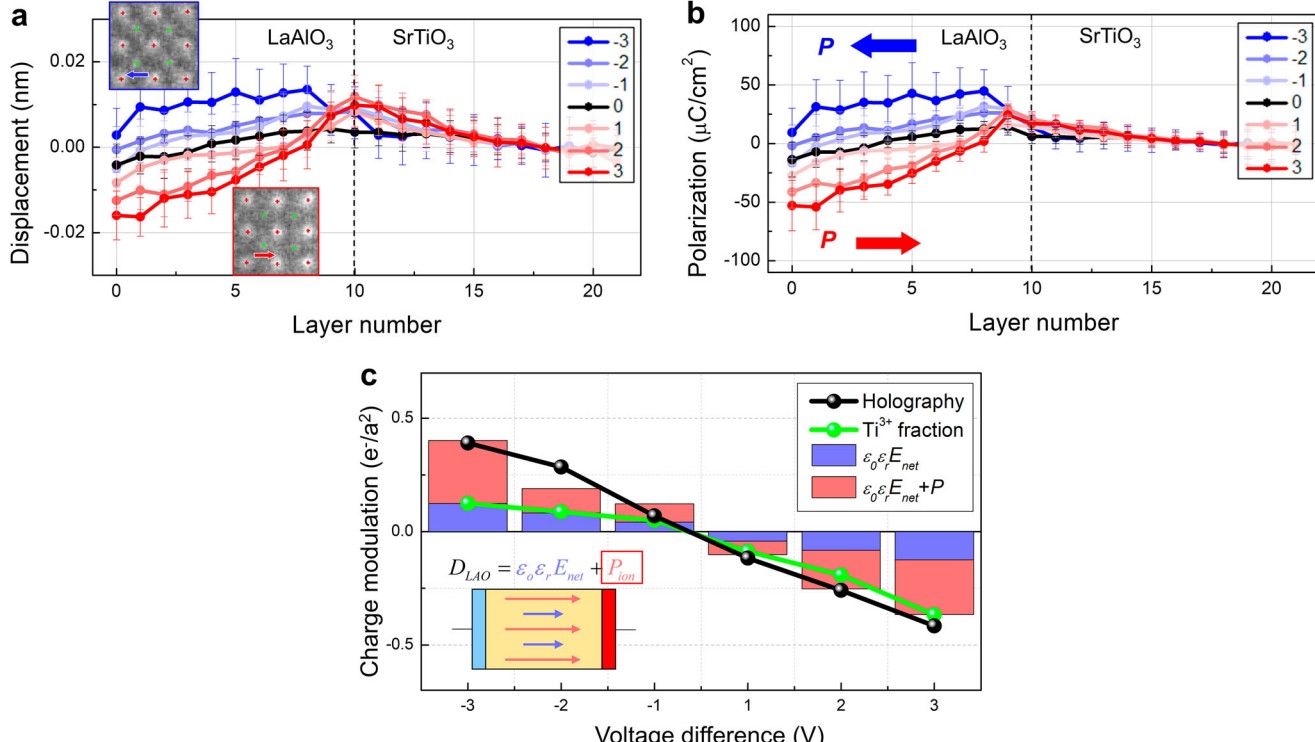

**Fig. 5 | Modulation of 2DEG density by field-induced ionic polarization and associated charges in LaAlO₃.** **a** Profiles of the displacement of B-site cation ($\delta_B$) measured across the LaAlO₃/SrTiO₃ heterostructure under applied voltages. **b** Profiles of the polarization ($P_i$) across the LaAlO₃/SrTiO₃ heterostructure under applied voltages. The out-of-plane component of polarization along the field direction was plotted. The polarization arises only in the LaAlO₃ and is switchable by the electric field. **c** Charge density modulating from the unbiased state plotted as a function of applied voltage. The values obtained by different methods are included; inline electron holography data (black dot) representing the net charge modulation, Ti³⁺ fraction determined from EELS Ti-L₂,₃ edge representing 2DEG modulation, the calculated dielectric charges ($\varepsilon_0\varepsilon_r E_a$, blue bar), and the dielectric charges ($\varepsilon_0\varepsilon_r E_a$) together with the measured polarization charges ($P_i$, red bar). The dielectric constant ($\varepsilon_r$) of 20 and 300 was used for LaAlO₃ and SrTiO₃, respectively. The modulation of net charge density is accounted for well by considering both dielectric charges ($\varepsilon_0\varepsilon_r E_a$) and polarization charge ($P_i$) of LaAlO₃.

These electrons are measured by inline electron holography. We note that not all incremental electrons measured by the inline electron holography are itinerant but only a portion of charges contribute to the transport and the rest are either localized in the Ti 3$d$ orbitals or trapped by defects[42].

For comparison of the two different types of measured charge modulation, the incremental Ti³⁺ state determined from the EELS data (green data points in Fig. 5c) was plotted together with the incremental net charges extracted from the holography. The incremental Ti³⁺ state at each voltage was determined by integrating the Ti³⁺ state up to four unit cells and subtracting the unbiased result. The density of the Ti³⁺ state at unbiased conditions is 0.15 e/$a^2$, which is comparable to the itinerant 2DEG density measured by transport characterization (Supplementary Fig. S2b). This value increases to 0.52 e/$a^2$ at + 3 V, resulting in an increment of 0.37 e/$a^2$. This value corresponds to 88% of the incremental net charges of 0.42 e/$a^2$ measured from the holography data. When negative voltages were applied, the density of Ti³⁺ state decreases with the voltage, albeit in a less pronounced manner, exhibiting a distinct asymmetry in its response to the applied field. It seems that the Ti³⁺ state density, corresponding to the density of 2DEG, does not diminish below a certain threshold but rather reaches saturation, thereby maintaining the stability of the polar field.

To verify the modulation of 2DEG by the $P_i$ induced in LaAlO₃, we constructed 2 × 2 (LaAlO₃)₉/(SrTiO₃)₅ slabs by using the atomic coordinates derived from the STEM images taken under biasing conditions. While the cation positions (La, Al, Sr, and Ti) were directly obtained from the STEM images, the oxygen positions were inferred, assuming that their displacement is twice that of the displacement of B-site cations[47–51]. We created three supercells corresponding to 0 V, − 1 V, and

+1 V conditions (Fig. 6). Subsequently, for the given ionic polarization $P_i$ of LaAlO₃ in each supercell only SrTiO₃ layers are allowed to relax to compensate for the polarization charge originating from the LaAlO₃ layer. Initially, at an unbiased condition (0 V), 2DEG formed in the SrTiO₃ layers, with a total density of 0.42 e/$a^2$, in the absence of the ionic polarization in LaAlO₃. Notably, post-relaxation, the SrTiO₃ layers showed significant distortion, aligning with the observed localized 2DEG (Supplementary Fig. S15), which agrees with the experimental findings. Upon introducing downward and upward ionic polarization in the LaAlO₃ (corresponding to +1 and −1 V, respectively), we observed changes in the 2DEG density; an increase to 0.47 e/$a^2$ for downward polarization and a decrease to 0.36 e/$a^2$ for upward polarization.

Given the thickness of the LaAlO₃/SrTiO₃ heterostructure and the applied voltage, the resulting electric field is approximately in the order of 10⁻⁶ V/cm, which is high enough to potentially trigger the electromigration of V_O. However, the observed activation of polar distortion as a response to counter the electric field, rather than electromigration, likely stems from its lower activation energy. It has been shown that V_O forms spontaneously at the LaAlO₃/SrTiO₃ surface when the thickness of LaAlO₃ exceeds the $t_c$ for 2DEG formation (4 u.c.) as the formation energy is negative[19]. However, as one moves inward from the surface of the LaAlO₃ layers, the formation energy of V_O rapidly transitions to positive values. This implies a higher energy barrier for the inward electromigration of V_O compared to that for polar distortion. When a stronger electric field is applied, it appears that the electromigration of V_O might be activated, followed by the initial polar distortion response, which is evidenced by a hysteresis effect in the I-V curve. In light of these observations, we suggest that the absence of V_O electromigration under specific experimental

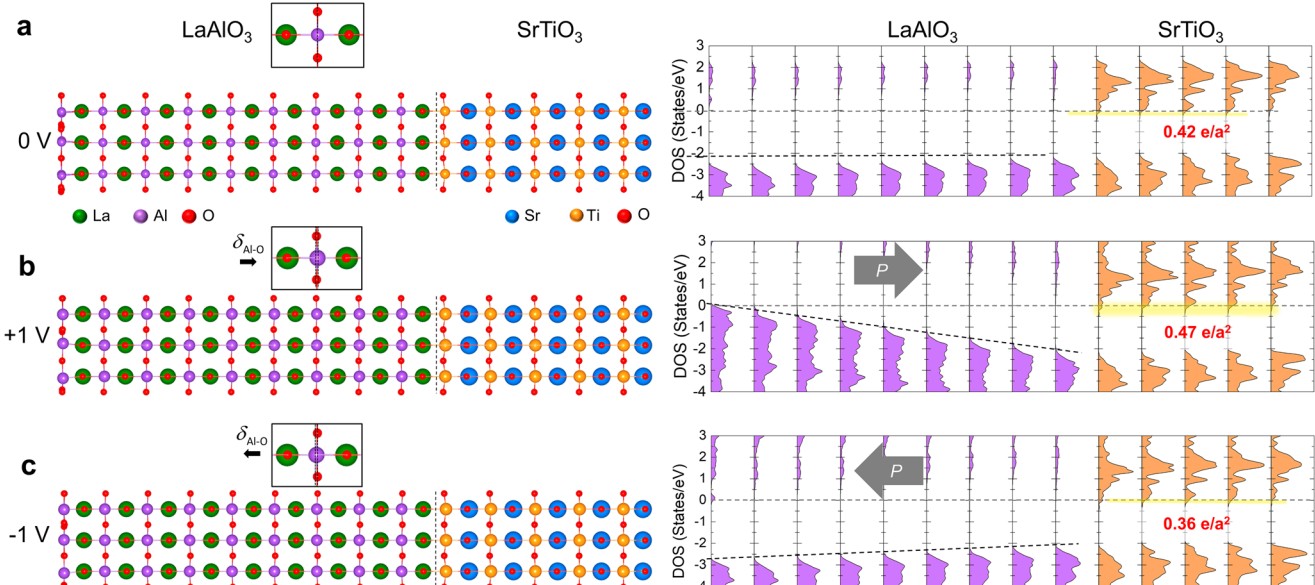

**Fig. 6 | DFT calculation rationalizing the 2DEG modulation by the field-induced polarization in LaAlO₃. a** Structural model ($2 \times 2$ $(LaAlO_3)_9/(SrTiO_3)_5$ slab) without ionic polarization ($P_i^o$) in the LaAlO₃ layer and layer-by-layer DOS of each TiO₂ and AlO₂ layer. $V_O$ was introduced to the LaAlO₃ surface to take account of the spontaneous formation of $V_O$ which acts as a source for 2DEG. Layer-by-layer DOS with (**b**) downward ($P_i^-$) and (**c**) upward polarization ($P_i^+$) in LaAlO₃. The charge density of 2DEG obtained by integrating the DOS below the Fermi level is estimated to be 0.42 e/$a^2$, 0.47 e/$a^2$ and 0.36 e/$a^2$ for $P_i^o$, $P_i^-$, and $P_i^+$, respectively.

conditions in this study is not an anomaly of the experimental setup but an intrinsic characteristic of the LaAlO₃/SrTiO₃ system.

## Discussion

In summary, comprehensive in-situ electron microscopy characterization under external bias visualized the field-induced modulation of 2DEG at the oxide heterointerface. Further, the multiple information gained by applying various STEM-based techniques is used to unambiguously resolve the origin of the modulation of 2DEG. In the particular case of the LaAlO₃ layer, the electromigration of $V_O$ present in the LaAlO₃ surface region is not activated. Instead, the strong polar distortion of the LaAlO₃ layer, which exhibits a large but volatile ionic polarization, is the main cause of the 2DEG modulation.

Our research marks a novel and practically important method for manipulating the polar distortion in LaAlO₃ epitaxial films. While earlier studies have established that different polar phases of LaAlO₃ films can be stabilized on a nonpolar substrate, influenced by variables such as film thickness, misfit strain, and interface orientation, our findings introduce electric field application as an additional method to control this distortion. This unique attribute of LaAlO₃ thin films, particularly when interface interactions with the substrate and electrode are finely tuned, opens up exciting possibilities in the field of nanoelectronics. It allows for the field-induced control of local charges, which can be pivotal in the development and function of various nanoelectronic devices.

## Methods
### Sample growth
SrRuO₃, LaAlO₃, and SrTiO₃ thin films were grown epitaxially on a TiO₂ terminated Nb:SrTiO₃ (001) substrate using pulsed laser deposition with in-situ reflection high-energy electron diffraction (RHEED) monitoring. To achieve a TiO₂-terminated substrate, as-received Nb:SrTiO₃ substrates were etched in a buffered HF (BHF) for 30 sec and annealed at 900 °C for 6 h. We first grew a 3 u.c. of LaAlO₃ epitaxial layer followed by a 15 u.c. of SrTiO₃ layer on Nb:SrTiO₃ (001) substrate. The thicknesses of LaAlO₃ and SrTiO₃ layers were controlled by monitoring RHEED oscillations. We implemented BHF etching to make the surface of the SrTiO₃ layer in SrTiO₃/LaAlO₃/Nb:SrTiO₃ heterostructure TiO₂-

termination. Then, a 10 u.c. thick LaAlO₃ film followed by a 20 nm thick SrRuO₃ layer was deposited on the top of SrTiO₃/ LaAlO₃/Nb:SrTiO₃ (001) heterostructures. During the LaAlO₃, SrTiO₃, and SrRuO₃ film growth, the temperature was kept at 750 °C, 750 °C, and 680 °C, respectively. The oxygen partial pressure was 0.015 mbar, 0.1 mbar, and 0.12 mbar, respectively.

The cross-sectional samples for in-situ (S)TEM biasing experiments were prepared on a MEMS chip via Ga⁺ ion beam milling at an accelerating voltage from 30 kV down to 5 kV using a dual-beam focused ion beam system (FIB, Helios 450F1, Thermo Fisher Scientific).

### Charge density analysis by using inline electron holography
Focal series bright-field TEM images were acquired for phase reconstruction by inline electron holography under applied voltages. A total 7 images were obtained for each data set, with the defocus step of 750 nm, where the maximum defocus is 2250 nm. The sample was tilted at a small angle to minimize dynamic diffraction, and an energy filter was applied with an energy slit of 10 eV to remove the inelastically scattered electrons[41]. The reconstruction has been done iteratively by using full-resolution wave reconstruction (FRWR) software[31]. The total number of iterations for each reconstruction was 1000. The phase images were reconstructed and converted into a map representing the projected electrostatic potential, based on the assumption of the phase-object approximation for non-magnetic materials. The charge-density map was obtained from the potential data by implementing Poisson's equation using a Laplacian image filter.

### Thickness measurement of TEM specimen by EELS log-ratio method
The thickness of the TEM specimen was estimated by using the EELS log-ratio method following the equation,

$$\frac{t}{\lambda} = \ln\left(\frac{I_{tot}}{I_{ZLP}}\right), \tag{2}$$

where the $I_{tot}$ is the total intensity including the zero-loss peak and plasmon intensity, the $I_{ZLP}$ is the intensity of the zero-loss peak, the $\lambda$ is

the inelastic mean free path and the $t$ is the thickness of TEM specimen. The mean free path of each material is estimated based on the Iakoubovskii model[56]. The resulting $t/\lambda$ profile of the TEM specimen showed no significant thickness gradient. The average thickness of the TEM specimens prepared for in-situ biasing experiments calculated from $t/\lambda$ was 106.5 nm, where the estimated $\lambda$ is 112.06 nm and 116.06 nm for SrTiO$_3$ and LaAlO$_3$, respectively. Considering the small difference $\lambda$ for each layer, the thickness is assumed to be constant throughout the entire field-of-view.

### EELS data acquisition

Aberration-corrected STEM (JEM-ARM300CF, JEOL) equipped with an energy filter (Gatan Quantum ER965) was used for EELS data acquisition. EELS line scan data across the LaAlO$_3$/SrTiO$_3$ interface were recorded in the energy range of 445 ~ 548 eV (for Ti-L$_{2,3}$ and O-K edges). The energy dispersion and dwell time of individual sets were 0.05 eV/pixel and 0.2 s/pixel, respectively. When the dwell time is set to 0.2 s, although the signal-to-noise ratio is not high enough to obtain a reliable signal, the artifacts from specimen drift and the electron beam damage on the specimen could be minimized. To enhance the signal-to-noise ratio, several data sets were collected from adjacent areas with a small dwell time, assuming the material properties are the same in the whole interface along the in-plane direction. After data acquisition, the data sets were spatially aligned in the out-of-plane direction of the specimen by using simultaneously obtained HAADF profiles and averaged to reduce the noise.

### Analysis of Ti valence state

The energy loss near edge structure (ELNES) was used for quantitative analysis of the valence state of Ti ions. The valence state of Ti ions in SrTiO$_3$ is 4 + without extra electrons in the 3$d$ orbital. When the excessive electrons are confined at the Ti 3$d$ orbital, the density of states in which the core electrons can be excited is reduced. As a result, the fine structure of the core-loss spectra is changed. To quantitatively analyze such differences, Ti$^{3+}$ and Ti$^{4+}$ reference spectra were collected from LaTiO$_3$ and SrTiO$_3$, respectively. Every experimental EEL spectra were deconvoluted by using multiple linear least square fitting (MLLS) which assumes that the original spectrum is a linear combination of reference spectra with a certain fraction. Although the MLLS fitting provides quantitative numbers of the valence state at each probe position, one should note that it has a limitation; it does not consider dynamic scattering and channeling effect which might spatially blur the resulting valence state profiles.

### Polarization measurement using HAADF STEM images

For high-angle annular dark-field (HAADF) imaging the detection angle in the range of 68–280 mrad was used. The convergence semi-angle of the focused probe was 23 mrad. The scan distortion was corrected through the image correlation with a 90-degree rotated image acquired right after one image was taken. The information from the fast scan direction in each image is used as a reference for correcting the distortion along the slow scan direction. The correction process continues iteratively until the difference between the two images is converged[57].

In a HAADF STEM image, the positions of individual atomic columns were determined by measuring the center-of-mass (CoM) of the column intensity via iterative methods. Then, using the position of the A-site cations (Sr for SrTiO$_3$ and La for LaAlO$_3$) the center of each unit cell was determined. From the center of each unit cell, the displacements of B-site cation ($\delta_B$) were calculated. The contrast of oxygen was too weak to determine the position of oxygen columns in the HAADF images. The displacement of the center of oxygen octahedral cage ($\delta_O$) from the center

of A-site cations was estimated from the $\delta_B$ by defining the ratio, $\kappa$, where

$$\delta_O = \kappa \delta_{Ti}.$$

$\kappa$ was assumed to be −2 in this study. Using the $\delta_B$ and $\delta_O$, the ionic polarization ($P_i$) was calculated by the following equation,

$$P = \frac{1}{V}\sum_i \delta_i Z_i = \frac{1}{V}(\delta_B Z_B + 3\delta_O Z_O) = \frac{1}{V}\delta_B(Z_B + 3\kappa Z_O)), \qquad (3)$$

where $Z_i$ is the Born effective charge of $i$ atom[52]. In this study, we used the following effective charges of: $Z_{La} = 4.45$; $Z_{Al} = 2.92$; $Z_{O1} = -2.48$ (LaO layer); $Z_{O2} = -2.44$ (AlO$_2$ layer); $Z_{Sr} = 2.56$; $Z_{Ti} = 7.42$; $Z_{O1} = -5.89$ (SrO layer); $Z_{O2} = -2.03$ (TiO$_2$ layer).

### DFT calculation

The first-principle DFT calculations were performed using the generalized gradient approximation Perdew-Burke-Ernzerhof (GGA-PBE-sol) exchange-correlation functionals[58] and the projector-augmented wave (PAW) method[59] with a plane wave basis as implemented in the Vienna ab initio simulation package (VASP) code[60]. The plane waves were included up to a kinetic energy cutoff of 450 eV. All calculations were converged in energy to $10^{-5}$ eV/cell.

To investigate the modulation of 2DEG by the ionic polarization in LaAlO$_3$, we employed $2 \times 2$ (LaAlO$_3$)$_9$/(SrTiO$_3$)$_5$ slab with an $n$-type interface that consists of LaO/TiO$_2$ layers. One oxygen vacancy was introduced to the surface of LaAlO$_3$ which acts as the source for 2DEG at the LaAlO$_3$/SrTiO$_3$ interface. The $2 \times 2$ (LaAlO$_3$)$_9$/(SrTiO$_3$)$_5$ slabs were constructed by using the atomic coordinates determined from the STEM images under applied voltages. While the position of cations (La, Al, Sr, and Ti) was directly accessed from the STEM images, the position of oxygen was determined based on the assumption that the displacement of oxygen is twice the displacement of B-site cation[47–51]. We constructed three supercells using the atomic coordinates determined from the STEM images recorded under 0 V, −1 V, and +1 V. The in-plane lattice constant of the $2 \times 2$ in-plane LaAlO$_3$/SrTiO$_3$ slab was fixed to the relaxed lattice parameter of $2 \times 2$ $a_{STO}$ ($a_{STO} = 3.907$ Å), which is very close to the experimental lattice constant of 3.905 Å. Subsequently, for the given ionic polarization of LaAlO$_3$, only the SrTiO$_3$ layers are allowed to relax to compensate the polarization charges from the LaAlO$_3$ layer until the forces were reduced to less than $5 \times 10^{-2}$ eV/Å. For the Brillouin-zone integration, Γ-centered $4 \times 4 \times 1$ k-point meshes were used. Initially, the total density of 2DEG in the SrTiO$_3$ layers is 0.42 e/$a^2$ at 0 V in the absence of the ionic polarization in LaAlO$_3$. We note that the SrTiO$_3$ layers are distorted significantly after relaxation, which is associated with the accommodation of localized 2DEG[19,53,54]. With the introduction of the downward and upward ionic polarization in the LaAlO$_3$, which was measured at the applied voltage of + 1 V and − 1 V, respectively, the 2DEG density increases to 0.46 e/$a^2$ and decreases to 0.34 e/$a^2$, respectively.

### Reporting summary

Further information on research design is available in the Nature Portfolio Reporting Summary linked to this article.

## Data availability

The authors declare that all relevant data supporting the key findings of this study are available within the article. All other raw data generated during the current study are available from the corresponding authors upon request.

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

## Acknowledgements

This work was supported by the Samsung Research Funding & Incubation Center of Samsung Electronics under Project Number SRFC-MA1702-01 (S.H.O. and J.L.) and partly by the National Research Foundation of Korea (NRF) funded by the Korea government (MSIT) (No. NRF-2020R1A2C2101735), Creative Materials Discovery Program (NRF-2019M3D1A1078296), and, the KENTECH Research Grant (KRG2022-01-019) (S.H.O.). The first-principle calculations were performed using the facilities of the Joint Supercomputer Center of the Russian Academy of Sciences (JSCC RAS). J.L. acknowledges the support of an NRF grant funded by the Korean government (NRF-2018R1A2B6004394). The TEM work at Sungkyunkwan University (SKKU) was supported by the Advanced Facility Center for Quantum Technology and the TEM work at the Korea Institute of Energy Technology (KENTECH) was supported by the Center for Shared Research Facilities (S.H.O). This research is funded by the Gordon and Betty Moore Foundation's EPiQS Initiative, grant GBMF9065 to C.B.E. and Vannevar Bush Faculty Fellowship (N00014-20-1-2844 (C.-B.E.)). Transport measurement at the University of Wisconsin–Madison was supported by the US Department of Energy (DOE), Office of Science, Office of Basic Energy Sciences (BES), under award number DE-FG02-06ER46327 (C.-B.E.).

## Author contributions

S.H.O. conceived the project. J.S. conducted all in-situ TEM biasing experiments and data analysis under the supervision of S.H.O., H.L., K.E. and K.L. prepared the LAO/STO 2DEG devices under the supervision of C.-B.E., T.M. and J. B. conducted DFT calculation under supervision of J.L. All authors contributed to interpretation of data and visualization of results. J.S. and S.H.O. wrote the manuscript.

## Competing interests

The authors declare no competing interests.
