## [Peer Review File · Nature Communications]

Field-induced modulation of two-dimensional electron gas at LaAlO₃/SrTiO₃ interface by polar distortion of LaAlO₃REVIEWER COMMENTS

Reviewer #1 (Remarks to the Author):

Combining several advanced transmission electron microscopy-based techniques (inline electron holography, EELS, scanning TEM), the manuscript provides experimental evidence on the effect of an applied external bias on the band alignment, the carrier density of the 2DEG at the LaAlO₃/SrTiO₃(001) interface which is shown to be coupled to a switchable ferroelectric-like polarization of the LaAlO₃ lattice. Additionally, DFT calculations are performed to prove a difference in the carrier density at the interface for positive and negative bias. However, only a few details are given on the latter, e.g. whether a dipole correction was used for this asymmetric setup. Can the authors also clarify how the geometries under applied bias were determined: on one hand it is mentioned that the experimental ionic displacements are used, but also that the system was fully relaxed. The experiment provides only information on the B cation displacement, but not on oxygen. It is not clear what is the reasoning behind choosing the shift of oxygen to be twice the one of the cation. Moreover, while it is noted that the oxygen displacement is in opposite direction, Fig. 4a rather indicates shifts of anions and cations in the same direction. In particular from Fig. 5 the ferroelectric distortion in the LaAlO₃ part is hardly visible and much more pronounced in the STO layers. The authors are encouraged to quantify the relaxation pattern in both the LAO and STO part besides the carrier density for the cases with and without applied bias. Since the calculation is performed in a (2x2) unit cell, it would be good to assess the role of octahedral tilts and rotations.

Moreover, the metallic SRO may play a significant role on the band alignment and possibly also on the presence of oxygen vacancies in the top LAO layer, as previous studies have shown for metallic contacts. This effect should be addressed.

Some of the relevant literature should be complemented: in general, the ferroelectric distortion in LaAlO₃ to counteract the internal electric field of the polar film and its significant effect on the carrier density has been reported previously e.g. in Phys. Rev. Lett. 102, 107602 (2009). While this is not the first study on the effect of external bias on LaAlO₃/SrTiO₃(001) and some of the reported effects are not surprising, it is the combination of methods and interrelation of effects that renders new insight. In particular the demonstration of a switchable ferroelectric distortion in LaAlO₃ and the lack of field induced oxygen migration are important results. Since the field-induced ferroelectric distortion is a central finding, a better quantification of this effect is necessary. Concerning the lack of electromigration the authors should address if this is due to the particular conditions applied in the setup and if there are boundaries for their observations. Two possible applications are mentioned "nanoelectronics devices and high-capacity energy storage systems", but the authors should elaborate more on the implications of their findings.

Reviewer #2 (Remarks to the Author):

The Manuscript describes an experimental microscopic study, using atomic resolution transmission electron microscopy, of the field effect in top-gated (SrRuO₃ top electrode) LAO/STO system.

The bottom electrode is a Nb-doped STO, separated by 3uc of insulating LAO from the 15uc/10uc LAO/STO heterostructure.

During the experiment the field is applied between the top SrRuO₃ electrode and the Nb-STO single crystal, held at the ground.

The main claim, as reflected in the title of the manuscript, is that carriers are modulated at the LAO/STO interface as result of "Ferroelectric-like" polar distortions in the LAO epitaxial film, and not by the oxygen-vacancies electromigration as inferred from other studies.

I think that the main claim is valid.

I have however some important remarks, which require substantial changes, from my point of view, in the main text and in the title.

- The Authors talk about ferroelectric-like distortion and ferroelectric switching in several parts of the paper.

However, according to the experiment, there is no change in the local electronic and structural properties of LAO and of Ti valence after application of the negative and positive bias. i.e. there is no remanent hysteretic polarization. In other words the dipole created with a bias applied are volatile, thus describing the phenomena as due to Ferroelectric-like polar distortion is basically incorrect.

This is one of the main change I think it is needed in the Manuscript. In particular, I would suggest the Authors to modify the paper as following;

remove ferroelectric-like from the title,

remove ferroelectric-like switching behavior in the abstract (it is not a ferroelectric switching...)

remove everywhere in the paper as much as possible the reference to ferroelectric-like distortions, and replace it with polar distortions only, as the distortions observed are qualitatively (not quantitatively) similar to the distortions one could get in a proper ferroelectric (like the parent BTO) but are not result of the establishment of a ferroelectric order in the LAO.

- Another major remark is concerning the characterization of the sample. While EELS suggest that a 2DEG is indeed formed in the NbSTO/LAO(3uc)/STO/LAO/SRO, there is no transport characterizations of the sample or of a similar one (for example without the SRO layer). As for the realization of an LAO/STO 2DEG usually the high quality of the STO single crystal is crucial, it is not always possible to be sure that a 2DEG is formed if the 2DEG is composed by an STO film and not and STO single crystal.

I think that it is important to provide electrical transport evidences that indeed a 2DEG is realized. I am aware that technically it could be difficult, but certainly not impossible for example in a twin sample deposited on an insulating STO without an SRO capping.

-The change in the charge estimated from the potential profile and from ab-initio is giant. I mean a change of 0.4 e/unit cell area, is close to the total charge needed to solve polar catastrophe in the system. Since oxygen vacancies do not explain this change, I wonder where all this charge is. Moreover it would important also to evaluate the amount of charge (or even measure by a capacitance measurements or using an electrometer) expected for the system and compare with the measured one.

-On the other hand, the change in Ti valence is much smaller compared to 0.4 e/areal unit cell. Indeed, it would correspond to a change of Ti valence from, let say, 4+ to 3.5, i.e. 50% Ti would be in a Ti³⁺ configuration. I think this is a discrepancy to be discussed.

- As last related remark, I think that the observed changes in Ti-valence is much more reasonable respect the estimated charge modulations from the potential profile, as usually the carrier density in top-gated LAO/STO do not change by an amount equal to $1-2 \cdot 10^{14} \text{ cm}^{-2}$.

Few minor remarks

-ref. 27, missing article number

-Even if obvious, it is important to state that the measurements were carried out at room temperature

Reviewer #3 (Remarks to the Author):

The manuscript by J. Seo et al. reports an analysis of the field-induced 2DEG modulation at the LAO/STO interface by means of in-situ inline electron holography technique. The authors claim that such modulation under applied electric field is due to a polar distortion in the LAO layer, which adds additional polarization charges at the 2DEG at LAO/STO interface. They rule out that this behavior could be caused by the oxygen vacancy migration by performing EELS analysis of the interface while external bias is applied.

Firstly, the work presents a novel study of significance for the field of oxides in order to clarify the long-standing discussion on the origin of the 2DEG modulation under electric field, so it could be very interesting and worth considering for publication in Nature

Communications. The manuscript is well written, figures are clear, the problem is sufficiently introduced and the story is accessible to a broad audience in the oxide community.

However, there are several aspects of the work that should be addressed to support the author's conclusions prior to any publication:

1) My main concern about the manuscript is the lack of evidence for the existence of a 2DEG at the LAO/STO interface considering the complexity of the heterostructure (SrRuO₃/LaAlO₃/SrTiO₃/LaAlO₃/Nb:SrTiO₃) studied. Very few research groups have managed to obtain a 2DEG between LAO and STO, when STO is in thin film form (*below I indicate the groups I know to the best of my knowledge), so I consider that it is of vital importance for the paper to demonstrate the existence of the 2DEG by measuring the transport properties in the heterostructure. As the present heterostructure does not allow to carry out the transport measurements due to the conductive substrate, I suggest the authors to prove the existence of 2DEG in a similar heterostructure but with STO as substrate, i.e., (LaAlO₃/SrTiO₃/LaAlO₃/SrTiO₃ (substrate)).

2) In my opinion, it is sufficiently demonstrated in the manuscript by EELS measurements that the modulation in the 2DEG is not driven by oxygen movement at the top LAO/STO interface, however it may simply be due to charge motion due to the considerable electric field applied during the experiments (despite the insertion of 3 u.c. of LAO acting as charge blocking). In my opinion, clarifying the existence of a 2DEG, will support that such modulation indeed comes from the modulation of the 2DEG confined charges at the LAO/STO interface.

3) In Fig. 1 b) (and also in Fig. 1 c)), it can be seen a qualitative change in the potential at the 3 u.c. LAO layer when -3V or 3V is applied. How the authors explain this fact? Detailed EELS profile focused on the STO (14 u.c.)/LAO (3 u.c) interface would help to clarify whether this can be due to the existence of a small amount of oxygen vacancies at this interface (a slight increase of Ti³⁺ fraction can be also seen in Fig. 3 d) at this interface for -3 V).

Overall, I would suggest that this work needs some major modifications before being considered to be published on Nature Communications.

*P. Brinks et al., Applied Physics Letters 98, 242904 (2011), C. W. Bark et al., PNAS 108 (12), 4720 (2011), Z. Huang et al., PRB 90, 125156 (2014), F. Gunkel et al., Nanoscale 7, 1013 (2015), J. Huang et al., Physica Status Solidi 17, 2200272 (2023).

REVIEWER COMMENTS

Reviewer #1 (Remarks to the Author):

Combining several advanced transmission electron microscopy-based techniques (inline electron holography, EELS, scanning TEM), the manuscript provides experimental evidence on the effect of an applied external bias on the band alignment, the carrier density of the 2DEG at the LaAlO₃/SrTiO₃ (001) interface which is shown to be coupled to a switchable ferroelectric-like polarization of the LaAlO₃ lattice.

Additionally, DFT calculations are performed to prove a difference in the carrier density at the interface for positive and negative bias. However, only a few details are given on the latter, e.g. whether a dipole correction was used for this asymmetric setup. Can the authors also clarify how the geometries under applied bias were determined: on one hand it is mentioned that the experimental ionic displacements are used, but also that the system was fully relaxed.

Reply: We agree with the reviewer's comment pointing out the lack of detailed information on the DFT calculations. Dipole correction was not used for this asymmetric setup. This is because a sufficient vacuum of about 18 Å was taken into consideration for asymmetric slabs. For such a large vacuum, the interaction between periodic images becomes less significant, as the influence of neighboring images diminishes with distance. In such cases, it has been known that the dipole correction may indeed become less crucial for obtaining accurate results.

The method we used to determine the geometries under applied bias is as follows. First, we constructed 2x2 (LaAlO₃)₉/(SrTiO₃)₅ slabs by using the atomic coordinates determined from the STEM images under applied bias. While the position of cations (La, Al, Sr and Ti) was directly accessible from the STEM images, the position of oxygen was determined based on the assumption that the displacement of oxygen is twice the displacement of B-site cation (the reasoning behind this is provided in the reply to the next comment). We constructed three supercells using the atomic coordinates determined from the STEM images recorded under 0 V, -1 V and +1 V. Subsequently, for the given ionic polarization of LaAlO₃ in each supercell only SrTiO₃ layers are allowed to relax to compensate the polarization charge originating from the LaAlO₃ layer until the forces were reduced to less than 5×10⁻² eV/Å. Initially, 2DEG was formed at the SrTiO₃ layers at unbiased state (0 V), of which total density was 0.42 e/a² in the absence of the ionic polarization in LaAlO₃. We note that the SrTiO₃ layers are distorted severely after relaxation, which is associated with the accommodation of localized 2DEG. With the introduction of the downward and upward ionic polarization measured at the applied voltage of +1 and -1 V, respectively, in the LaAlO₃, the 2DEG density increases to 0.47 e/a² and decreases to 0.36 e/a², respectively (refer to revised Fig. 6).

We have revised the "DFT calculations" section in Materials and Methods by providing detailed information, which reads as follows.

Revised manuscript: The first-principle DFT calculations were performed using the generalized gradient approximation Perdew-Burke-Ernzerhof (GGA-PBEsol) exchange-correlation functionals⁶² and the projector-augmented wave (PAW) method⁶³ with a plane wave basis as implemented in the Vienna *ab initio* simulation package (VASP) code⁶⁴. The plane waves were included up to a kinetic energy cutoff of 450 eV. All calculations were

converged in energy to 10^{-5} eV/cell.

To investigate the modulation of 2DEG by the ionic polarization in LaAlO_3 , we employed 2×2 $(\text{LaAlO}_3)_9/(\text{SrTiO}_3)_5$ slab with an n -type interface that consists of LaO/TiO_2 layers. One oxygen vacancy was introduced to the surface of SrTiO_3 which acts as source for 2DEG at the $\text{LaAlO}_3/\text{STO}$ interface. The 2×2 $(\text{LaAlO}_3)_9/(\text{SrTiO}_3)_5$ slabs were constructed by using the atomic coordinates determined from the STEM images under applied bias. While the position of cations (La, Al, Sr and Ti) was directly accessed from the STEM images, the position of oxygen was determined based on the assumption that the displacement of oxygen is twice the displacement of B-site cation [Cantoni et al., *Adv. Mater.* **24**, 29 (2012); Jia, C.-L. et al. *Nat Mater* **6**, 64–69 (2007); Li et al., *Adv. Func. Mater.* **29**, 1906655 (2019); Wang et al., *ACS. Appl. Mater.* **10**, 1374–1382 (2018); Spurgeon et al., *Nat. Comm.* **6**, 7735 (2015); Wang, S. et al. *ACS Appl Mater Interfaces* **10**, 1374–1382 (2018)]. We constructed three supercells using the atomic coordinates determined from the STEM images recorded under 0 V, -1 V and +1 V. The in-plane lattice constant of the 2×2 in-plane $\text{LaAlO}_3/\text{SrTiO}_3$ slab was fixed to the relaxed lattice parameter of 2×2 a_{STO} ($a_{\text{STO}} = 3.907$ Å), which is very close to the experimental lattice constant of 3.905 Å. Subsequently, for the given ionic polarization of LaAlO_3 , only the SrTiO_3 layers are allowed to relax to compensate the polarization charges from the LaAlO_3 layer until the forces were reduced to less than 5×10^{-2} eV/Å. For the Brillouin-zone integration, Γ -centered $4 \times 4 \times 1$ k-point meshes were used. Initially, the total density of 2DEG in the SrTiO_3 layers is 0.42 e/a^2 at 0 V in the absence of the ionic polarization in LaAlO_3 . We note that the SrTiO_3 layers are distorted significantly after relaxation, which is associated with the accommodation of localized 2DEG [Song et al, *Adv. Sci.*, **8**, 2002073 (2021); Pauli et al. *Phys. Rev. Lett.*, **106**, 036101 (2011); Jia et al. *Phys. Rev. B* **79**, 081405R (2009)]. With the introduction of the downward and upward ionic polarization in the LaAlO_3 , which was measured at the applied voltage of +1 V and -1 V, respectively, the 2DEG density increases to 0.46 e/a^2 and decreases to 0.34 e/a^2 , respectively.

The experiment provides only information on the B cation displacement, but not on oxygen. It is not clear what is the reasoning behind choosing the shift of oxygen to be twice the one of the cation.

Reply: In the HAADF STEM images which we used for the measurement of ionic polarization, the contrast of oxygen columns is too weak to measure their displacements. The lack of oxygen contrast in HAADF STEM image for polarization measurement has been discussed in several papers. Other STEM imaging techniques such as ABF and iDPC would be applied to measure the oxygen displacement but could not be utilized in this work as the thickness of our TEM samples was optimized for in-situ biasing experiments (In order for the TEM specimen to represent the bulk characteristics, the inner undamaged region should be thick enough compared to the damaged surface layers by FIB).

As we understand that the displacement of oxygen is not ignorable and thus must be included in the calculation of polarization, we have gone through the literature for reasonable treatment of oxygen displacement. We paid particular attention to the literature reporting the experimental measurement of the oxygen displacement in $\text{LaAlO}_3/\text{SrTiO}_3$ systems [Cantoni et al., *Adv. Mater.* **24**, 29 (2012); Jia, C.-L. et al. *Nat Mater* **6**, 64–69 (2007); Li et al., *Adv. Func. Mater.* **29**, 1906655 (2019); Wang et al., *ACS. Appl. Mater.* **10**, 1374–1382 (2018); Spurgeon et al., *Nat. Comm.* **6**, 7735 (2015); Wang, S. et al. *ACS Appl Mater Interfaces* **10**, 1374–1382 (2018)]. We arrived at a conclusion that treating the oxygen displacement as

twice the B cation displacement is highly consistent with the literature.

As to the charges of oxygen and cations for the calculation of polarization, we used the effective charges reported in the literature [Behtash et al., *Phys. Chem. Chem. Phys.* **18**, 6831 (2016)].

We added the reasoning behind choosing the displacement of oxygen and cited relevant literature in the revised manuscript, which reads:

Revised manuscript: The P_i was determined on the u.c. basis by measuring the displacement of the B-site ions (δ_B) from the center of A-site sublattice in the HAADF STEM images (Fig. 4a). The contrast of oxygen columns was too weak to measure their displacements (δ_O) directly from the HAADF STEM images. As the displacement of oxygen is not ignorable and thus must be included in the calculation of P_i , we have gone through the literature for reasonable treatment of oxygen displacement [Cantoni et al., *Adv. Mater.* **24**, 29 (2012); Jia, C.-L. et al. *Nat Mater* **6**, 64–69 (2007); Li et al., *Adv. Func. Mater.* **29**, 1906655 (2019); Wang et al., *ACS. Appl. Mater.* **10**, 1374–1382 (2018); Spurgeon et al., *Nat. Comm.* **6**, 7735 (2015); Wang, S. et al. *ACS Appl Mater Interfaces* **10**, 1374–1382 (2018)]. For the given displacement of the B-site cation, it was assumed that the displacement of oxygen is twice as large in the opposite direction. As to the charges of oxygen and cations for the calculation of polarization, we used the effective charges reported in the literature [Behtash et al., *Phys. Chem. Chem. Phys.* **18**, 6831 (2016)].

Moreover, while it is noted that the oxygen displacement is in opposite direction, Fig. 4a rather indicates shifts of anions and cations in the same direction.

Reply: The unit cell model illustrated in Fig. 4a was prepared simply to define two different types of displacement one has to consider, i.e., the displacement of B-site cations and oxygen ions from the centrosymmetric position of A-site cations. This schematic illustration was prepared without considering the actual displacements in LaAlO₃. Having the reviewer's comment, to avoid the confusion and make it in line with the observation, we revised the model to conform to the displacements observed in the LaAlO₃ layers.

In particular from Fig. 5 the ferroelectric distortion in the LaAlO₃ part is hardly visible and much more pronounced in the STO layers. The authors are encouraged to quantify the relaxation pattern in both the LAO and STO part besides the carrier density for the cases with and without applied bias. Since the calculation is performed in a (2x2) unit cell, it would

be good to assess the role of octahedral tilts and rotations.

Reply: We appreciate the reviewer's comment, which led us to consider the polar distortion in SrTiO₃—an important phenomenon that was not discussed in the manuscript. As the reviewer pointed out, the polar distortion of the interfacial SrTiO₃ layers is pronounced, which results from the accommodation of 2DEG [Song et al, *Adv. Sci.*, **8**, 2002073 (2021), Pauli, et al. *Phys. Rev. Lett.* **106**, 036101 (2011)]. The electrons injected from the LaAlO₃ surface to the interfacial SrTiO₃ layers are confined by band bending. The gradient in the band-bending region results in an electric field in the opposite direction to that in the film. This causes negative buckling of the SrTiO₃ layers once the 2DEG is formed as also seen in our previous work [Song et al, *Adv. Sci.*, **8**, 2002073 (2021)]. As depicted in Figure R1, both our experimental measurements and DFT calculations consistently demonstrate that this phenomenon is universal.

Figure R1. Polar distortion in the LaAlO₃/SrTiO₃ (001) systems where 2DEG is formed at the interface. δ_B and δ_O denotes the displacement of B-site cation and oxygen, respectively. (a) DFT calculation results reported in literature (Song et al, *Adv. Sci.* **8**, 2002073 (2021)). (b) Experimental measurement using STEM images and (c) DFT calculation results from this study.

On the other hand, the reason why the polar distortion in LaAlO₃ is less pronounced compared to that in SrTiO₃ is that we used the polar distortion measured at the smallest applied voltages of +1 and -1 V. Following the reviewer's suggestion, we have included the entire set of displacement patterns across the LaAlO₃/SrTiO₃ interface obtained at various applied voltages in Fig. 4.

As to the octahedral tilts, first we would like to point out that it has been known that the octahedral tilts and polar distortion in LaAlO₃/SrTiO₃ (001) system are not compatible but compete with each other [Gazquez, et al., *Phys. Rev. Lett.* **119**, 106102 (2017); Min et al, *Sci. Adv.* **7**, eabe9053 (2021)]. When the polar distortion becomes dominant in the LaAlO₃ layer due to the applied field as in the present study or by the internal polar field below the critical thickness for 2DEG formation, the octahedral tilts disappear. In the present study, we cannot determine the octahedral tilts experimentally in the LaAlO₃ and SrTiO₃ layers using HAADF STEM images due to the weak oxygen contrast. In the DFT calculations, as we used the experimentally measured atomic positions and relaxed only the SrTiO₃ layers, we only can assess the octahedral tiles in the SrTiO₃ layers after relaxation. We could not observe the octahedral tilts except the polar distortion in the SrTiO₃ layers, which is associated with the accommodation of 2DEG.

In the revised manuscript, we added the detailed discussion on the relaxation pattern including the polar distortion in SrTiO₃ and octahedral tilts.

Revised manuscript: We note that the SrTiO₃ layers beneath the LaAlO₃/SrTiO₃ interface exhibits a noticeable polar distortion (blue pixels in Figs. 4b, c and Fig. 5a), which is related with the accommodation of 2DEG (Song et al, *Adv. Sci.* **8**, 2002073 (2021); Pauli et al. *Phys. Rev. Lett.* **106**, 036101 (2011); Jia et al. *Phys. Rev.* **B79**, 081405R (2009)). The electrons injected from the V_O at LaAlO₃ surface to the interface are confined to the near-interface SrTiO₃ region by band bending. The gradient of band-bending in this region results in the electric field which causes the polar distortion of SrTiO₃ layers (Supplementary Fig. S14) as also seen in our previous work (Song et al, *Adv. Sci.* **8**, 2002073 (2021)).

As to the octahedral tilts in the LaAlO₃/SrTiO₃ (001) system, we cannot determine the octahedral tilts experimentally in the LaAlO₃ and SrTiO₃ layers using HAADF STEM images due to the weak oxygen contrast. It has been known that the octahedral tilts and polar distortion are not compatible but compete with each other in the LaAlO₃/SrTiO₃ (001) system [Gazquez, et al., *Phys. Rev. Lett.* **119**, 106102 (2017); Min et al, *Sci. Adv.* **7**, eabe9053 (2021)]. When the polar distortion becomes dominant in the LaAlO₃ layers due to the applied field as in the present study or by the internal polar field below the t_c for 2DEG formation [Pentcheva and Prof. Pickett, *Phys. Rev. Lett.* **102**, 107602 (2009)], the octahedral tilts may not coexist with the polar distortion but tend to disappear. Indeed, we could not observe the octahedral tilts except the polar distortion in the SrTiO₃ layers in the DFT calculation.

Moreover, the metallic SRO may play a significant role on the band alignment and possibly also on the presence of oxygen vacancies in the top LAO layer, as previous studies have shown for metallic contacts. This effect should be addressed.

Reply: The work function of SrRuO₃ and LaAlO₃ is known as 5.2 eV and 4.8 eV in the literature [Zhao et al., *Sci. Rep.* **5**, 9680 (2015); Gao et al., *Adv. Mater. Interfaces* **5**, 1701565 (2018)], respectively. This work function difference (0.4 eV) is expected to introduce a Schottky barrier at the SrRuO₃/LaAlO₃ interface. As the reviewer noted, the band alignment and contact at the SrRuO₃/LaAlO₃ interface can be influenced by charged defects such as oxygen vacancies. The charged oxygen vacancies can result in the pinning of Fermi level and/or lowering of the barrier height (Andr  et al., *Adv. Electron Mater.* **6**, 1900808 (2019); Chrysler et al., *Phys. Rev. Mater.* **5**, 104603 (2021)), affecting the transport across the interface.

The analysis of the I-V curves measured during in-situ STEM biasing experiments (Supplementary Fig. S6) can provide a clue to the conduction mechanism and thereby the characteristic of the SrRuO₃/LaAlO₃ contact. Choosing the most reliable the I-V curve from multiple TEM samples which can represent the device characteristics (the black curve in Fig. S6, Fig. S12a), we analyzed the curve according to the various transport mechanisms. Among various fitting trials of this I-V curve, the best match is found for the log I-log V plot, which exhibits a well-defined "space-charge-limited conduction (SCLS)" behavior controlled by trap sites. At low voltages, the current shows a clear Ohmic (slope, $m = 1$) behavior, followed by transition to a trap-filling-limited ($m = I + 1$, where $I = 6.08$ and 4.33 for positive and negative voltages, respectively) behaviors. At higher voltages, almost all traps are filled and thus trap-free, space-charge-limited ($m = 2$) conduction behavior governs the transport. Based on this I-V characteristics, we can describe the SrRuO₃/LaAlO₃ interface as an Ohmic contact with the trap sites originating from oxygen vacancies. We added a new supplementary figure (Fig. S12) and related description in the revised manuscripts, which reads as follows.

Figure R2. I-V curve of the LaAlO₃/SrTiO₃ heterostructure TEM sample showing the smallest leakage current (black curve in Fig. S6). (a) Linear scale and (b) log-scale plot in the voltage ranges from -0.8 to 0.8 V. The slopes (m) of three different regimes were determined by linear fitting. The transition from ohmic ($m=1$) to space-charge limited current is identified. At low voltages, the current shows a clear Ohmic (slope, $m=1$) behavior, followed by transition to a trap-filling-limited ($m=1+l$, where $l=6.08$ and 4.33 for positive and negative voltages, respectively) behaviors. At higher voltages, almost all traps are filled and thus trap-free, space-charge-limited ($m=2$) conduction behavior governs the transport.

Revised manuscript: The spatially confined V_O at the LaAlO₃ surface can influence the band alignment and contact at the interface with electrode (here SrRuO₃). The work function of SrRuO₃ and LaAlO₃ is 5.2 eV and 4.8 eV, respectively (Zhao et al., *Sci. Rep.* **5**, 9680 (2015); Gao et al., *Adv. Mater. Interfaces* **5**, 1701565 (2018)). This work function difference (0.4 eV) is expected to introduce a Schottky barrier at the interface. The charged V_O can result in the pinning of Fermi level and/or lowering of the barrier height [Andr  et al., *Adv. Electron Mater.* **6**, 1900808 (2019); Chrysler et al., *Phys. Rev. Mater.* **5**, 104603 (2021)], affecting the transport across the interface. The analysis of the I-V curves measured during in-situ STEM biasing experiments (Supplementary Fig. 12) provides a clue to the conduction mechanism and thereby the characteristic of the SrRuO₃/LaAlO₃ contact. Choosing the most reliable the I-V curve from multiple TEM samples which can represent the device characteristics (the black curve in Supplementary Fig. 6b, Fig. 12a), we analyzed the curve according to the various transport mechanisms. Among various approaches to fitting this I-V curve, the best match was obtained by the log I-log V plot. This plot clearly demonstrates a space-charge-limited conduction behavior, which is governed by trap sites. Based on this I-V characteristics, the SrRuO₃/LaAlO₃ interface can be characterized as an Ohmic contact, with the trap sites primarily originating from V_O .

Some of the relevant literature should be complemented: in general, the ferroelectric distortion in LaAlO₃ to counteract the internal electric field of the polar film and its significant effect on the carrier density has been reported previously e.g. in *Phys. Rev. Lett.* **102**, 107602 (2009).

Reply: We appreciate the reviewer for catching out missing a highly relevant literature to our work. We are well aware of the pioneering work by Prof. Pentcheva and Prof. Pickett reporting the polar distortion of LaAlO₃ which counteracts the internal electric field and makes significant effect on the carrier density. We cited the literature, addressing its relevance to our work, as follows.

Revised manuscript: When the polar distortion becomes dominant in the LaAlO₃ layer due to the applied field as in the present study or by the internal polar field below the t_c for 2DEG formation [Pentcheva and Pickett, *Phys. Rev. Lett.* **102**, 107602 (2009)],...

While this is not the first study on the effect of external bias on LaAlO₃/SrTiO₃(001) and some of the reported effects are not surprising, it is the combination of methods and interrelation of effects that renders new insight. In particular the demonstration of a switchable ferroelectric distortion in LaAlO₃ and the lack of field induced oxygen migration are important results. Since the field-induced ferroelectric distortion is a central finding, a better quantification of this effect is necessary.

Reply: We appreciate the reviewer for evaluating the central finding of our work high. We agree with the reviewer that more quantitative description of the field-induced polar distortion of LaAlO₃ can strengthen our finding. We thereby presented the layer-by-layer polar distortion and ionic polarization measured at each applied voltage in Fig. 5a and b, respectively (The original Fig. 4 was split into two figures (Fig. 4 and Fig. 5) in the revised manuscript with adding more detailed information). To show the trend of the evolution of polar distortion and ionic polarization with applied voltage more clearly, we added the polarization results obtained at other applied voltages in Fig. 4 and 5. Figure

Fig. 4. Ionic polarization induced in LaAlO₃/SrTiO₃ heterostructure by electric field. **a**, Atomic model of ABO₃ perovskite oxide depicting the ionic displacements. The displacement of B-site cation and oxygen from the center of A-site cation sublattice is defined as δ_B and δ_O , respectively. **c**, Polarization (P_i) map of the LaAlO₃/SrTiO₃ heterostructure determined using HAADF STEM recorded at unbiased condition (0 V). δ_B was measured directly from the HAADF STEM image but δ_O was approximated as two times the δ_B in opposite direction. For the calculation of polarization the effective charges reported in the literature [Behtash et al., *Phys. Chem. Chem. Phys.* **18**, 6831 (2016)] are used (refer to Methods). Only the out-of-plane component of polarization along the field direction was displayed in the map. **c**, P_i maps obtained under applied voltage conditions from +/-1 V to +/- 3V. The P_i is induced in the LaAlO₃ layer along the direction of applied field. The P_i signal appearing in the SrTiO₃ is the buckling associated with the accommodation of 2DEG [Song et al, *Adv. Sci.* **8**, 2002073 (2021); Pauli et al. *Phys. Rev. Lett.* **106**, 036101 (2011); Jia et al. *Phys. Rev. B* **79**, 081405R (2009)].

Fig. 5. Modulation of 2DEG density by field-induced ionic polarization and associated charges in LaAlO₃. **a**, Profiles of the displacement of B-site cation (δ_B) measured across the LaAlO₃/SrTiO₃ heterostructure under applied voltages. **b**, Profiles of the polarization (P_i) across the LaAlO₃/SrTiO₃ heterostructure under applied voltages. The out-of-plane component of polarization along the field direction was plotted. The polarization arises only in the LaAlO₃ and is switchable by electric field. **c**, Charge density modulating from the unbiased state plotted as a function of applied voltage. The values obtained by different methods are included; inline electron holography data (black dot) representing the net charge modulation, Ti³⁺ fraction determined from EELS Ti-L_{2,3} edge representing 2DEG modulation, the calculated dielectric charges ($\epsilon_0\epsilon_r E_a$, blue bar), and the dielectric charges ($\epsilon_0\epsilon_r E_a$) together with the measured polarization charges (P_i , red bar). The dielectric constant (ϵ_r) of 20 and 300 was used for LaAlO₃ and SrTiO₃, respectively. The modulation of net charge density is accounted for well by considering both dielectric charges ($\epsilon_0\epsilon_r E_a$) and polarization charge (P_i) of LaAlO₃.

Concerning the lack of electromigration the authors should address if this is due to the particular conditions applied in the setup and if there are boundaries for their observations.

Reply: Considering the film thickness of LaAlO₃/SrTiO₃ heterostructure and applied voltage, the induced electric field is in the order of 10^{-6} V/cm, which is certainly high enough to activate the electromigration of V_O. The reason why the polar distortion is activated to cancel the electric field rather than electromigration is probably due to the lower activation energy. It has been shown that V_O forms spontaneously on the surface of LaAlO₃/SrTiO₃ above the critical thickness for 2DEG formation (4 unit cell) as the formation energy is negative. The formation energy increases steeply to positive values as it moves inward the LaAlO₃ layer from the surface. For this reason, the energy barrier for the electromigration of V_O inward the LaAlO₃ layers must be higher than that of the polar distortion. When a higher electric field is applied, the electromigration of V_O is likely to be activated after the initial activation of polar distortion with exhibiting a hysteresis in I-V curve. In this context, we propose that the lack of electromigration is not related to the particular conditions applied in the setup but intrinsic behavior of the system. The discussion was added to the revised manuscript.

Revised manuscript: Given the thickness of the LaAlO₃/SrTiO₃ heterostructure and the applied voltage, the resulting electric field is approximately in the order of 10^{-6} V/cm, which is

high enough to potentially trigger the electromigration of V_O . However, the observed activation of polar distortion as a response to counter the electric field, rather than electromigration, likely stems from its lower activation energy. It has been shown that V_O forms spontaneously at the $\text{LaAlO}_3/\text{SrTiO}_3$ surface when the thickness of LaAlO_3 exceeds the t_c for 2DEG formation (4 u.c.) as the formation energy is negative [Song, K. *et al. Advanced Science* **8**, 2002073 (2021)]. However, as one moves inward from the surface of the LaAlO_3 layers, the formation energy of V_O rapidly transitions to positive values. This implies a higher energy barrier for the inward electromigration of V_O compared to that for polar distortion. When a stronger electric field is applied, it appears that electromigration of V_O might be activated, followed by the initial polar distortion response, which is evidenced by a hysteresis effect in the I-V curve. In light of these observations, we suggest that the absence of V_O electromigration under the specific experimental conditions is not an anomaly of the setup but an intrinsic characteristic of the $\text{LaAlO}_3/\text{SrTiO}_3$ system.

Two possible applications are mentioned “nanoelectronics devices and high-capacity energy storage systems”, but the authors should elaborate more on the implications of their findings.

Reply: Following the reviewer’s advice, we elaborated more on the implications of our finding as it follows.

Revised manuscript: Our research marks a novel and practically important method for manipulating the polar distortion in LaAlO_3 epitaxial films. While earlier studies have established that different polar phases of LaAlO_3 films can be stabilized on a nonpolar substrate, influenced by variables such as film thickness, misfit strain, and interface orientation, our findings introduce electric field application as an additional method to control this distortion. This unique attribute of LaAlO_3 thin films, particularly when interface interactions with the substrate and electrode are finely tuned, opens up exciting possibilities in the field of nanoelectronics. It allows for the field-induced control of local charges, which can be pivotal in the development and function of various nanoelectronic devices.

Reviewer #2 (Remarks to the Author):

The Manuscript describes an experimental microscopic study, using atomic resolution transmission electron microscopy, of the field effect in top-gated (SrRuO₃ top electrode) LAO/STO system.

The bottom electrode is a Nb-doped STO, separated by 3uc of insulating LAO from the 15uc/10uc LAO/STO heterostructure.

During the experiment the field is applied between the top SrRuO₃ electrode and the Nb-STO single crystal, held at the ground.

The main claim, as reflected in the title of the manuscript, is that carriers are modulated at the LAO/STO interface as result of "Ferroelectric-like" polar distortions in the LAO epitaxial film, and not by the oxygen-vacancies electromigration as inferred from other studies.

I think that the main claim is valid.

I have however some important remarks, which require substantial changes, from my point of view, in the main text and in the title.

- The Authors talk about ferroelectric-like distortion and ferroelectric switching in several parts of the paper.

However, according to the experiment, there is no change in the local electronic and structural properties of LAO and of Ti valence after application of the negative and positive bias. i.e. there is no remanent hysteretic polarization. In other words the dipole created with a bias applied are volatile, thus describing the phenomena as due to Ferroelectric-like polar distortion is basically incorrect.

This is one of the main change I think it is needed in the Manuscript. In particular, I would suggest the Authors to modify the paper as following;

remove ferroelectric-like from the title,

remove ferroelectric-like switching behavior in the abstract (it is not a ferroelectric switching...)

remove everywhere in the paper as much as possible the reference to ferroelectric-like distortions, and replace it with polar distortions only, as the distortions observed are qualitatively (not quantitatively) similar to the distortions one could get in a proper ferroelectric (like the parent BTO) but are not result of the establishment of a ferroelectric order in the LAO.

Reply: We respect the reviewer's knowledgeable comment on the field-induced polar distortion in the LaAlO₃. Having the comment, we agree that "ferroelectric-like" is not an appropriate wording to describe the observed field-induced polar distortion in the LaAlO₃. Accepting the reviewer's suggestion, we removed "ferroelectric-like" through the manuscript including the title.

- Another major remark is concerning the characterization of the sample. While EELS suggest that a 2DEG is indeed formed in the NbSTO/LAO(3uc)/STO/LAO/SRO, there is no transport characterizations of the sample or of a similar one (for example without the SRO layer). As for the realization of an LAO/STO 2DEG usually the high quality of the STO single crystal is crucial, it is not always possible to be sure that a 2DEG is formed if the 2DEG is

composed by an STO film and not and STO single crystal.

I think that it is important to provide electrical transport evidences that indeed a 2DEG is realized. I am aware that technically it could be difficult, but certainly not impossible for example in a twin sample deposited on an insulating STO without an SRO capping.

Reply: We understand the reviewer's concern of using SrTiO₃ film instead of SrTiO₃ single crystal substrate for the realization of 2DEG. As demonstrated by inline electron holography charge density maps and EELS Ti-L_{2,3} edge, the 2DEG was indeed formed at LaAlO₃/SrTiO₃ interface even though we used 20 u.c.-thick STO film. Our growth team has been working on the improvement of the crystal quality of SrTiO₃ film to realize the 2DEG at the interface of LaAlO₃ through either the optimization of PLD growth condition or the modification of growth method [Eom et. al., *Sci. Adv.* **7**, eabh1284, (2021)]. Nevertheless, following the reviewer's comment, we prepared a model film structure on undoped SrTiO₃ substrate without SrRuO₃ top electrode for transport measurement. The interface charge density, resistance and mobility data obtained from the transport characterization confirms the formation of 2DEG at LaAlO₃/SrTiO₃ interface (Figure R3, included as Supplementary Fig. S2). Compared to the 2DEG formed at a standard LaAlO₃ (10 uc)/ SrTiO₃ substrate sample (Figure R4, included as Supplementary Fig. S3), the density of 2DEG formed at the model device structure was lower and the resistance was higher probably due to the unavoidable crystal imperfections in SrTiO₃ film. This lower carrier density of 2DEG is also consistent with our EELS results measured in the unbiased state ($< 0.5e^-/a^2$). We have included the electrical transport measurements of both the model device and standard substrate sample in the Supplementary information as Fig. S2 and S3, respectively, and briefly described the results in the revised manuscript.

Revised manuscript: Considering that the formation of 2DEG at LaAlO₃/SrTiO₃ interface is contingent upon the crystalline quality of SrTiO₃, the application of SrTiO₃ film might pose limitations on the formation of 2DEG. To investigate the electrical transport of the 10 u.c. LAO/15 u.c. SrTiO₃ (001) interface used in this study, we fabricated a similar structure on an undoped SrTiO₃ substrate, but without the SrRuO₃ top electrode. The transport characterization, including interface charge density, resistance, and mobility measurements, confirmed the formation of 2DEG at the LaAlO₃/SrTiO₃ interface, as shown in Supplementary Fig. S2. When compared to a standard LaAlO₃ (10 uc)/SrTiO₃ substrate sample (Supplementary Fig. S3), the 2DEG density in our model device structure was found to be lower, and its resistance higher. This disparity is attributable to unavoidable crystal imperfections in the SrTiO₃ film.

Figure R3. Transport measurement of a 10 unit cell thick LaAlO₃ grown on 15 unit cell thick SrTiO₃ (001) grown on LaAlO₃ buffered SrTiO₃ (001) substrate. The 10 u.c.-LaAlO₃/15 u.c.-SrTiO₃/3 u.c.-LaAlO₃ model structure was grown on undoped SrTiO₃ (001) substrate without SrRuO₃ top electrode for the electrical transport measurement. (a) Resistance, (b) carrier density, and (c) mobility.

Figure R4. Transport measurement of a standard 10 unit cell thick LaAlO_3 grown on a SrTiO_3 (001) substrate. (a) Resistance, (b) carrier density, and (c) mobility.

-The change in the charge estimated from the potential profile and from ab-initio is giant. I mean a change of 0.4 e/unit cell area, is close to the total charge needed to solve polar catastrophe in the system. Since oxygen vacancies do not explain this change, I wonder where all this charge is. Moreover it would be important also to evaluate the amount of charge (or even measure by capacitance measurements or using an electrometer) expected for the system and compare with the measured one.

Reply: We appreciate the reviewer for this comment. The source of the incremental charges at the $\text{LaAlO}_3/\text{SrTiO}_3$ interface by the applied field should have been discussed in more detail. Regarding the source of the incremental charges, one should pay attention to the current flowing through the $\text{LaAlO}_3/\text{SrTiO}_3$ interface due to the electronic transport when the two electrodes are biased against each other through top-gating (refer to the I-V curves in Fig. R5 and included as Supplementary Fig. S12 in the revised manuscript).

The analysis of the I-V curves measured during in-situ STEM biasing experiments (Supplementary Fig. S6) can provide a clue to the conduction mechanism. Choosing the most reliable the I-V curve from multiple TEM samples which can represent the device characteristics (the black curve in Fig. S6, Fig. S12a), we analyzed the curve according to the various transport mechanisms. Among various fitting trials of this I-V curve, the best match was found for the log I-log V plot, which exhibits a well-defined “space-charge-limited conduction (SCLS)” behavior controlled by trap sites. At low voltages, the current shows a clear Ohmic (slope, $m = 1$) behavior, followed by transition to a trap-filling-limited ($m = l + 1$, where $l = 6.08$ and 4.33 for positive and negative voltages, respectively) behaviors. At higher voltages, almost all traps are filled and thus trap-free, space-charge-limited ($m = 2$) conduction behavior governs the transport. Based on this I-V characteristics, we can describe the $\text{SrRuO}_3/\text{LaAlO}_3$ interface as an Ohmic contact with the trap sites originating from oxygen vacancies. We added a new Supplementary Fig. S12 and related description in the revised manuscripts, which reads as follows.

Figure R5. I-V curve of the LaAlO₃/SrTiO₃ heterostructure TEM sample showing the smallest leakage current (black curve in Fig. S6). (a) Linear scale and (b) log-scale plot in the voltage ranges from -0.8 to 0.8 V. The slopes (m) of three different regimes were determined by linear fitting. The transition from Ohmic ($m=1$) to space-charge limited current is identified. At low voltages, the current shows a clear Ohmic (slope, $m = 1$) behavior, followed by transition to a trap-filling-limited ($m = l+1$, where $l = 6.08$ and 4.33 for positive and negative voltages, respectively) behaviors. At higher voltages, almost all traps are filled and thus trap-free, space-charge-limited ($m = 2$) conduction behavior governs the transport.

Revised manuscript: The spatially confined V_O at the LaAlO₃ surface can influence the band alignment and contact at the interface with electrode (here SrRuO₃). The work function of SrRuO₃ and LaAlO₃ is 5.2 eV and 4.8 eV, respectively. (Zhao et al., *Sci. Rep.* **5**, 9680 (2015); Gao et al., *Adv. Mater. Interfaces* **5**, 1701565 (2018)). This work function difference (0.4 eV) is expected to introduce a Schottky barrier at the interface. The charged V_O can result in the pinning of Fermi level and/or lowering of the barrier height [Andr  et al., *Adv. Electron Mater.* **6**, 1900808 (2019); Chrysler et al., *Phys. Rev. Mater.* **5**, 104603 (2021)], affecting the transport across the interface. The analysis of the I-V curves measured during in-situ STEM biasing experiments (Supplementary Fig. S12) provides a clue to the conduction mechanism and thereby the characteristic of the SrRuO₃/LaAlO₃ contact. Choosing the most reliable the I-V curve from multiple TEM samples which can represent the device characteristics (the black curve in Supplementary Fig. S6b, Fig. S12a), we analyzed the curve according to the various transport mechanisms. Among various approaches to fitting this I-V curve, the best match was obtained by the log I-log V plot. This plot clearly demonstrates a space-charge-limited conduction behavior, which is governed by trap sites. Based on this I-V characteristics, the SrRuO₃/LaAlO₃ interface can be characterized as an Ohmic contact, with the trap sites primarily originating from V_O

Reply: These electrons transporting between the electrodes are the major source of the incremental charges, leading to the modulation of the intrinsic 2DEG formed at the LAO/STO interface. These charges are attracted to or repulsed from the LaAlO₃/SrTiO₃ interface to compensate the bound charges arising in LaAlO₃ from both ordinary dielectric response and additional ionic polarization of LaAlO₃. The former, represented by blue bar in Fig. 5c, is calculated by using $\sigma = \frac{\epsilon_0 \epsilon_r}{t} V = \epsilon_0 \epsilon_r E$. The latter corresponds to the charges from the measured ionic polarization (P_i) in LaAlO₃. The summation of the two contributions ($D = \epsilon_0 \epsilon_r E + P_i$), represented by red bar in Fig. 5c, constitutes the net bound charges. The electrons transporting between the two electrodes are attracted to or repulsed from the LaAlO₃/SrTiO₃ interface to compensate the net bound charges. These electrons are

measured by inline electron holography. We note that not all incremental electrons measured by inline holography are itinerant but only a portion of charges contribute to the transport and the rest are either localized in the Ti 3d orbitals such as d_{xz} and d_{yz} or trapped by defects. [Song et al., *Nature Nanotechnology* **13**, 198-203, (2018)].

Revised manuscript: The electrons transporting between the two electrodes (SrRuO₃ and Nb:SrTiO₃) are the major source leading to the modulation of the 2DEG at the LaAlO₃/SrTiO₃ interface. These charges are attracted to or repulsed from the LaAlO₃/SrTiO₃ interface to compensate the bound charges arising in the LaAlO₃ from both ordinary dielectric response and additional ionic polarization of LaAlO₃. The former, represented by blue bar in Fig. 5c, is calculated by using $\epsilon_0\epsilon_r E_a$. The latter corresponds to the charges from the measured ionic polarization (P_i) in LaAlO₃. The summation of the two contributions ($\epsilon_0\epsilon_r E_a + P_i$), represented by red bar in Fig. 5c, constitutes the net bound charges. The electrons transporting between the two electrodes are attracted to or repulsed from the LaAlO₃/SrTiO₃ interface to compensate the net bound charges. These electrons are measured by inline electron holography. We note that not all incremental electrons measured by the inline electron holography are itinerant but only a portion of charges contribute to the transport and the rest are either localized in the Ti 3d orbitals such as d_{xz} and d_{yz} or trapped by defects. (Song et al., *Nature Nanotechnology* **13**, 198-203, (2018)).

-On the other hand, the change in Ti valence is much smaller compared to 0.4 e/areal unit cell. Indeed, it would correspond to a change of Ti valence from, let say, 4+ to 3.5, i.e. 50% Ti would be in a Ti³⁺ configuration. I think this is a discrepancy to be discussed.

Reply: As we already discussed in our previous publication [Song et al., *Nature Nanotechnology* **13**, 198-203, (2018)], the incremental charges detected by the Ti valence state (Ti³⁺) from EELS Ti-L_{2,3} is different from that measured by inline holography as these two techniques probe fundamentally different types of charges. The charge density measured by inline holography are the net charge density encompassing all charges, not only the itinerant 2DEG which contributes to the lateral transport but also localized or trapped charges at defects which do not contribute to the transport. On the other hand, the incremental charges detected by the change of Ti valence state by EELS Ti-L_{2,3} edge are the electrons occupying the Ti-3d orbitals. Therefore, the incremental 2DEG density assessed by the change of Ti valence state from EELS Ti-L_{2,3} edge is usually smaller than that measured by inline electron holography.

For comparison of the two different types of charge modulation, the incremental Ti³⁺ state determined from the EELS data was plotted together with the incremental net charges extracted from holography data in Fig. 5c. The incremental Ti³⁺ state at each voltage was determined by integrating the Ti³⁺ state up to four unit cells and subtracting the unbiased result. For example, the Ti³⁺ state of 0.15 e/a² at unbiased state increases to 0.52 e/a² at +3 V, resulting the increment of 0.37 e/a². This value corresponds to 88% of the increment of net charges of 0.42 e/a² measured from the holography data. We added the discussion of this point to the revised manuscript, which reads:

Revised manuscript: For comparison of the two different types of the measured charge modulation, the incremental Ti³⁺ state determined from the EELS data (green data points in Fig. 5c) was plotted together with the incremental net charges extracted from the holography. The incremental Ti³⁺ state at each voltage was determined by integrating the Ti³⁺ state up to four unit cells and subtracting the unbiased result. The density of Ti³⁺ state at

unbiased condition is $0.15 e/a^2$, which is comparable to the itinerant 2DEG density measured by transport characterization (Supplementary Fig. S2b). This value increases to $0.52 e/a^2$ at +3 V, resulting in the increment of $0.37 e/a^2$. This value corresponds to 88% of the incremental net charges of $0.42 e/a^2$ measured from the holography data. When negative voltages were applied, the density of Ti^{3+} state decreases with the voltage, albeit in a less pronounced manner, exhibiting a distinct asymmetry in its response to the applied field. It seems that the Ti^{3+} state density, corresponding to the density of 2DEG, does not diminish below a certain threshold but rather reaches saturation, thereby maintaining the stability of the polar field.

- As last related remark, I think that the observed changes in Ti-valence is much more reasonable respect the estimated charge modulations from the potential profile, as usually the carrier density in top-gated LAO/STO do not change by an amount equal to $1-2 \cdot 10^{14} \text{ cm}^{-2}$.

Reply: As we discussed for the previous comments, the Ti^{3+} state determined by EELS $Ti-L_{2,3}$ is more directly related with the modulation of the itinerant charges which we can attribute to 2DEG. In the case of the charge modulation from the potential profile it still encompasses the modulation of all charges, not only the itinerant 2DEG but also localized or trapped charges. In this context, when it comes to the modulation of 2DEG, we agree with the reviewer that Ti valence change is much more reasonable. We revised the related description in the manuscript to reflect the discussion and made necessary changes to the figures.

Few minor remarks

-ref. 27, missing article number

Reply: Thank you for catching out this incomplete citation of literature. We included the missing article number of ref. 27 (now ref. 26 in the revised manuscript).

-Even if obvious, it is important to state that the measurements were carried out at room temperature

Reply: We stated that all the measurements were carried out at room temperature.

Revised manuscript: The in-situ TEM biasing was performed by applying a DC voltage to the $SrRuO_3$ electrode, ranging from -3 V to +3 V in 1 V increments at room temperature.

Reviewer #3 (Remarks to the Author):

The manuscript by J. Seo et al. reports an analysis of the field-induced 2DEG modulation at the LAO/STO interface by means of in-situ inline electron holography technique. The authors claim that such modulation under applied electric field is due to a polar distortion in the LAO layer, which adds additional polarization charges at the 2DEG at LAO/STO interface. They rule out that this behavior could be caused by the oxygen vacancy migration by performing EELS analysis of the interface while external bias is applied.

Firstly, the work presents a novel study of significance for the field of oxides in order to clarify the long-standing discussion on the origin of the 2DEG modulation under electric field, so it could be very interesting and worth considering for publication in Nature Communications. The manuscript is well written, figures are clear, the problem is sufficiently introduced and the story is accessible to a broad audience in the oxide community. However, there are several aspects of the work that should be addressed to support the author's conclusions prior to any publication:

We appreciate the reviewer for the recognition of the novelty of our work.

1) My main concern about the manuscript is the lack of evidence for the existence of a 2DEG at the LAO/STO interface considering the complexity of the heterostructure (SrRuO₃/LaAlO₃/SrTiO₃/LaAlO₃/Nb:SrTiO₃) studied. Very few research groups have managed to obtain a 2DEG between LAO and STO, when STO is in thin film form(*below I indicate the groups I know to the best of my knowledge), so I consider that it is of vital importance for the paper to demonstrate the existence of the 2DEG by measuring the transport properties in the heterostructure. As the present heterostructure does not allow to carry out the transport measurements due to the conductive substrate, I suggest the authors to prove the existence of 2DEG in a similar heterostructure but with STO as substrate, i.e., (LaAlO₃/SrTiO₃/LaAlO₃/SrTiO₃ (substrate)).

Reply: We understand the reviewer's concern of using SrTiO₃ film instead of SrTiO₃ single crystal substrate for the realization of 2DEG. As demonstrated by inline electron holography charge density maps and EELS Ti-L_{2,3} edge, the 2DEG was indeed formed at LaAlO₃/SrTiO₃ interface even though we used 20 u.c.-thick STO film. Our growth team has been working on the improvement of the crystal quality of SrTiO₃ film to realize the 2DEG at the interface of LaAlO₃ through either the optimization of PLD growth condition or the modification of growth method [Eom et. al., *Sci. Adv.* **7**, eabh1284, (2021)]. Nevertheless, following the reviewer's comment, we prepared a model film structure on undoped SrTiO₃ substrate without SrRuO₃ top electrode for transport measurement. The interface charge density, resistance and mobility data obtained from the transport characterization confirms the formation of 2DEG at LaAlO₃/SrTiO₃ interface (Figure R3, included as Supplementary Fig. S2). Compared to the 2DEG formed at a standard LaAlO₃ (10 uc)/ SrTiO₃ substrate sample (Figure R4, included as Supplementary Fig. S3), the density of 2DEG formed at the model device structure was lower and the resistance was higher probably due to the unavoidable crystal imperfections in SrTiO₃ film. This lower carrier density of 2DEG is also consistent with our EELS results measured in the unbiased state (0.15 e⁻/a²). We have included the electrical transport measurements of both the model device and standard substrate sample in the Supplementary information as Fig. S2 and S3, respectively, and briefly described the results in the revised manuscript.

Revised manuscript: Considering that the formation of 2DEG at LaAlO₃/SrTiO₃ interface is

contingent upon the crystalline quality of SrTiO₃, the application of SrTiO₃ film might pose limitations on the formation of 2DEG. To investigate the electrical transport of the 10 u.c. LAO/15 u.c. SrTiO₃ (001) interface used in this study, we fabricated a similar structure on an undoped SrTiO₃ substrate, but without the SrRuO₃ top electrode. The transport characterization, including interface charge density, resistance, and mobility measurements, confirmed the formation of 2DEG at the LaAlO₃/SrTiO₃ interface, as shown in Supplementary Fig. S2. When compared to a standard LaAlO₃ (10 uc)/SrTiO₃ substrate sample (Supplementary Fig. S3), the 2DEG density in our model device structure was found to be lower, and its resistance higher. This disparity is attributable to unavoidable crystal imperfections in the SrTiO₃ film.

Figure R3. Transport measurement of a 10 unit cell thick LaAlO₃ grown on 15 unit cell thick SrTiO₃ (001) grown on LaAlO₃ buffered SrTiO₃ (001) substrate. The 10 u.c.-LaAlO₃/15 u.c.-SrTiO₃/3 u.c.-LaAlO₃ model structure was grown on undoped SrTiO₃ (001) substrate without SrRuO₃ top electrode for the electrical transport measurement. (a) Resistance, (b) carrier density, and (c) mobility.

Figure R4. Transport measurement of a standard 10 unit cell thick LaAlO₃ grown on a SrTiO₃ (001) substrate. (a) Resistance, (b) carrier density, and (c) mobility.

2) In my opinion, it is sufficiently demonstrated in the manuscript by EELS measurements that the modulation in the 2DEG is not driven by oxygen movement at the top LAO/STO interface, however it may simply due to charge motion due to the considerable electric field applied during the experiments (despite the insertion of 3 u.c. of LAO acting as charge blocking). In my opinion, clarifying the existence of a 2DEG, will support that such modulation indeed comes from the modulation of the 2DEG confined charges at the LAO/STO interface.

Reply: As shown in the reply to the previous comment, the 2DEG was indeed formed at the LaAlO₃/SrTiO₃ interface of our sample. As the reviewer pointed out correctly, when the two electrodes are biased against each other through top-gating, there is current flowing through

the LaAlO₃/SrTiO₃ interface due to the electronic transport (despite the insertion of 3 u.c. of LaAlO₃ acting as charge blocking). The electronic charges transporting between the electrodes via space-charge-limited conduction (SCLC) are the major source of the incremental charges, leading to the modulation of the intrinsic 2DEG formed at the LAO/STO interface. These charges are attracted to or repulsed from the LaAlO₃/SrTiO₃ interface to compensate the bound charges arising from the ionic polarization in the LaAlO₃.

Figure R5. I-V curve of the LaAlO₃/SrTiO₃ heterostructure TEM sample showing the smallest leakage current (black curve in Fig. S6). (a) Linear scale and (b) log-scale plot in the voltage ranges from -0.8 to 0.8 V. The slopes (m) of three different regimes were determined by linear fitting. The transition from Ohmic ($m=1$) to space-charge limited current is identified. At low voltages, the current shows a clear Ohmic (slope, $m = 1$) behavior, followed by transition to a trap-filling-limited ($m = l + 1$, where $l = 6.08$ and 4.33 for positive and negative voltages, respectively) behaviors. At higher voltages, almost all traps are filled and thus trap-free, space-charge-limited ($m = 2$) conduction behavior governs the transport.

As we discussed in our previous publication [Song et al., *Nature Nanotechnology* **13**, 198-203, (2018)], the incremental charges detected by the Ti valence state from EELS Ti-L_{2,3} is different from that measured by inline holography as these two techniques probe fundamentally different types of charges. The charge density from the potential profile obtained by inline holography corresponds to the net charge density encompassing all charges, not only the itinerant charges which contribute to the lateral transport but also localized or trapped charges at defects which do not contribute to the transport. On the other hand, the incremental charges detected by the change of Ti valence state by EELS Ti-L_{2,3} edge are the electrons occupying the Ti-3d orbitals. Therefore, the incremental 2DEG density assessed by the change of Ti valence state from EELS Ti-L_{2,3} edge is usually smaller than that measured by inline electron holography. We added the discussion of this point to the revised manuscript, which reads:

Revised manuscript: The electrons transporting between the two electrodes (SrRuO₃ and Nb:SrTiO₃) are the major source leading to the modulation of the 2DEG at the LaAlO₃/SrTiO₃ interface. These charges are attracted to or repulsed from the LaAlO₃/SrTiO₃ interface to compensate the bound charges arising in the LaAlO₃ from both ordinary dielectric response and additional ionic polarization of LaAlO₃. The former, represented by blue bar in Fig. 5c, is calculated by using $\epsilon_0 \epsilon_r E_a$. The latter corresponds to the charges from the measured ionic polarization (P_i) in LaAlO₃. The summation of the two contributions ($\epsilon_0 \epsilon_r E_a + P_i$), represented by red bar in Fig. 5c, constitutes the net bound charges. The electrons transporting between the two electrodes are attracted to or repulsed from the LaAlO₃/SrTiO₃ interface to compensate the net bound charges. These electrons are measured by inline electron

holography. We note that not all incremental electrons measured by the inline electron holography are itinerant but only a portion of charges contribute to the transport and the rest are either localized in the Ti 3d orbitals such as d_{xz} and d_{yz} or trapped by defects. (Song et al., *Nature Nanotechnology* **13**, 198-203, (2018)).

3) In Fig. 1 b) (and also in Fig. 1 c)), it can be seen a qualitative change in the potential at the 3 u.c. LAO layer when -3V or 3V is applied. How the authors explain this fact? Detailed EELS profile focused on the STO (14 u.c.)/LAO (3 u.c) interface would help to clarify whether this can be due to the existence a small amount of oxygen vacancies at this interface (a slight increase of Ti^{3+} fraction can be also seen in Fig. 3 d) at this interface for -3 V).

Reply: We appreciate the reviewer for careful examination of the potential profiles and EELS data near the $SrTiO_3$ (15 u.c.)/ $LaAlO_3$ (3 u.c.) interface and pointing out the change in the potential in the $LaAlO_3$ layer. We did not pay too much attention to such a thin $LaAlO_3$ layer since no significant change is observed in EELS O-K edge and Ti-L_{2,3} edge (refer to Fig. R6 below). The slight increase of Ti^{3+} fraction seen in Fig. 3d is only one data point, for which it is hard to assign a specific meaning. Further, the inline electron holography has a limited spatial resolution ~ 0.8 nm and lack of low frequency information [as discussed in the current manuscript and also in our paper, Seo et al., *Ultramicroscopy* **231**, 113236 (2021)], the interpretation of potential and charge data in the $LaAlO_3$ (3 u.c.) layer facing the two interfaces closely in ~ 1.1 nm is challenging. Although we cannot exclude a possibility of charge redistribution within the $LaAlO_3$ (3 u.c.) layer under the applied electric field, it is still challenging to elucidate this phenomenon with the techniques we employed in this study. We anticipate that employing more advanced techniques with a high spatial resolution and charge sensitivity such as electron ptychography can resolve this issue in future research.

Figure R6. EELS O-K and Ti-L_{2,3} edges obtained from the $SrTiO_3$ region close to the $SrTiO_3$ (15 u.c.)/ $LaAlO_3$ (3 u.c.) interface under the applied voltage of -3 V (blue) and +3 V (red). No significant change attributable to oxygen vacancies is observed in both spectra.

Overall, I would suggest that this work needs some major modifications before being considered to be published on Nature Communications.

Reply: Thanks to all your knowledgeable comments, we could improve the manuscript by strengthening the weak points. We hope that our replies and revision satisfy the reviewer.

*P. Brinks et al., Applied Physics Letters 98, 242904 (2011), C. W. Bark et al., PNAS 108 (12), 4720 (2011), Z. Huang et al., PRB 90, 125156 (2014), F. Gunkel et al., Nanoscale 7, 1013 (2015), J. Huang et al., Physica Status Solidi 17, 2200272 (2023).

Reply: We cited the references suggested by the reviewer.

REVIEWER COMMENTS

Reviewer #2 (Remarks to the Author):

I have read with interest the rebuttal letter and the changes made in the manuscript according to the reviewers remarks.

In general I appreciate the efforts made by the Authors to clear out their finding, in particular the use of Polar displacements, distortions and so on instead ferroelectric-like distortions which was somewhat misleading.

I also appreciate the new data which demonstrate that the 2DEG at the interface with the epitaxial STO has properties not far from standard LAO/STO 2DEG.

I am still not convinced about the amount of charge estimated from in-line holography and the discrepancy with EELS data.

Electrons are expected to go (either localized or not) into the STO-Ti 3d-states. Sr-related states are too far in energy. Thus, concerning STO, I think that the estimated charge measured by in-line holography and EELS should be similar.

The Authors believe that EELS is probing only itinerant electrons, which is not. Similarly to XAS, the Ti-L_{2,3} edge EELS spectra is related to the Valence of the Ti-ion. The valence is Ti³⁺ also (or mainly) when electrons are localized into Ti 3d states. Of course also conduction electrons contribute to a lowering of the average Ti-valence, but in general EELS and XAS are less sensitive to itinerant carriers and more to localized electrons.

Thus localized electrons in Ti 3d_{xz,yz}, or any other Ti-3d orbital, will give a contribution to the EELS which is distinctly different from empty Ti-sites. Indeed EELS can be used to estimate the amount of Ti³⁺.

Thus, either the "missing" electron counts are somewhere else, not in the STO and in Ti-related states, or there is a discrepancy between EELS and in-line holography measurements.

First I need the Authors to change the sentence : "...but that detected by the change of Ti valence state by EELS Ti-L_{2,3} edge are the itinerant electrons occupying the Ti-3d orbitals."

since this is not true.

Secondly, I need the Authors to reconsider the possible origin of the differences between in-holography and EELS.

Can it be that they are actually measuring also electrons trapped in LAO or somewhere else?

I believe that this is one crucial point which does not allow me to recommend publication at this stage.

Reviewer #3 (Remarks to the Author):

The authors convincingly address all my concerns about the manuscript so I recommend the paper for publication in Nature Communications.

REVIEWER COMMENTS

Reviewer #2 (Remarks to the Author):

I have read with interest the rebuttal letter and the changes made in the manuscript according to the reviewers remarks.

In general I appreciate the efforts made by the Authors to clear out their finding, in particular the use of Polar displacements, distortions and so on instead ferroelectric-like distortions which was somewhat misleading.

I also appreciate the new data which demonstrate that the 2DEG at the interface with the epitaxial STO has properties not far from standard LAO/STO 2DEG.

I am still not convinced about the amount of charge estimated from in-line holography and the discrepancy with EELS data.

Electrons are expected to go (either localized or not) into the STO-Ti 3d-states. Sr-related states are too far in energy. Thus, concerning STO, I think that the estimated charge measured by in-line holography and EELS should be similar.

The Authors believe that EELS is probing only itinerant electrons, which is not. Similarly to XAS, the Ti-L_{2,3} edge EELS spectra is related to the Valence of the Ti-ion. The valence is Ti³⁺ also (or mainly) when electrons are localized into Ti 3d states. Of course also conduction electrons contribute to a lowering of the average Ti-valence, but in general EELS and XAS are less sensitive to itinerant carriers and more to localized electrons. Thus localized electrons in Ti 3d_{xz,yz}, or any other Ti-3d orbital, will give a contribution to the EELS which is distinctly different from empty Ti-sites. Indeed EELS can be used to estimate the amount of Ti³⁺.

Thus, either the "missing" electron counts are somewhere else, not in the STO and in Ti-related states, or there is a discrepancy between EELS and in-line holography measurements.

First I need the Authors to change the sentence : "...but that detected by the change of Ti valence state by EELS Ti-L_{2,3} edge are the itinerant electrons occupying the Ti-3d orbitals."

since this is not true.

Reply: We sincerely appreciate the reviewer's insightful comments and advice concerning the interpretation of EELS data and its underlying fundamental principles. We concur with the reviewer that the Ti³⁺ state determined from the EELS Ti-L_{2,3} edges indicates that electrons are localized in the 3d states of Ti. However, we cannot ascertain whether these localized electrons are itinerant without analyzing their transport behaviors.

In response to this invaluable feedback, we have meticulously revisited our manuscript and made necessary revisions to improve the precision of the descriptions associated with this data.

Revised manuscript (page 10):

The field-induced modulation of the 2DEG can be assessed by analyzing changes in the fine structure of the EELS Ti-L_{2,3} edge in response to varying applied voltages⁴⁴ (Fig. 3, Supplementary Fig. S10). It is well-established that the 2DEG predominantly occupies empty Ti 3d orbitals, thereby reducing the valence state of Ti from 4+ to 3+. Specifically, at the LaAlO₃/SrTiO₃ (001) interface, the 2DEG preferentially fills the *d_{xy}* orbital within the *t_{2g}* band, the lowest energy state⁴³. Applying an electric field can alter the 2DEG density, either increasing to or decreasing the electrons localized in the Ti 3d states. These changes in the 2DEG density are reflected in the variation of the Ti³⁺ fraction with applied voltages, particularly observable through changes in the intensity of the e_g peaks at both Ti-L₃ and L₂ edges, as these electron states are sensitive to changes detected by EELS (Supplementary Fig. S10). Since the features of both Ti³⁺ and Ti⁴⁺ state co-exists in the EELS Ti-L_{2,3} edge to varying extents, the Ti³⁺ fraction has been quantified using multiple linear least squares (MLLS) fitting of the EELS Ti-L_{2,3} edge with reference spectra for Ti³⁺ and Ti⁴⁺ (Supplementary Fig. S11).

Secondly, I need the Authors to reconsider the possible origin of the differences between inline holography and EELS.

Can it be that they are actually measuring also electrons trapped in LAO or somewhere else?

I believe that this is one crucial point which does not allow me to recommend publication at this stage.

Reply: We appreciate the reviewer's insightful comment prompting us to re-evaluate the discrepancies observed between the EELS and inline holography data. We reconsidered several factors that could possibly contribute to these differences. Notably, the variation in spatial resolution between the two methods significantly affects the locality of the data obtained.

The EELS data were acquired in STEM mode using an Ångström-level electron probe, providing high spatial resolution. In contrast, inline holography data are collected in TEM bright-field mode, where the use of an objective aperture limits the spatial resolution to approximately 0.7 nm. This resolution constraint results in the integration of the charge density signal across the LaAlO₃/SrTiO₃ interface in inline holography measurements. Additionally, inline holography is sensitive to net charges, encompassing not only Ti³⁺ but also other local charges near the interface. Among the potential sources of the additional charges other than 2DEG included in the inline holography data, we considered the electron trapped in the defects present in the near-interface region of LaAlO₃. It is well known that that some extent of cation intermixing between LaAlO₃ and SrTiO₃ is unavoidable at the interface during the growth of LaAlO₃ on SrTiO₃. Considering this, it is likely that some additional electrons trapped near the interface region of LaAlO₃ are included in the inline

holography data.

Given these factors, the EELS data specifically detect Ti^{3+} states in the SrTiO_3 region, while inline holography captures the broader net charges across the somewhat blurred $\text{LaAlO}_3/\text{SrTiO}_3$ interface. We have added a brief discussion to the manuscript, addressing these critical points and the potential overestimation of charge density by inline holography due to these differences in technique and resolution.

Revised manuscript (page 9):

However, note that due to the finite spatial resolution of the technique (~ 0.7 nm) the change in the 2DEG density in the SrTiO_3 results in the change of the peaks across the $\text{LaAlO}_3/\text{SrTiO}_3$ interface (Fig. 2f, Supplementary Fig. S9f). For the quantitative measurement of charge modulation, therefore, the areal integration has been conducted over an extended distance across the interface and the corresponding value at unbiased state was subtracted. The total amount of additional charge density increased (decreased) at +3 V (-3 V) with respect to the unbiased state is measured to be about $0.41 e/a^2$, where a is the lattice parameter. We note that the charge density measured by inline electron holography are the density of net charges encompassing all charges, not only the 2DEG which contributes to the transport but also other localized or trapped charges which do not contribute to the transport. Considering that some extent of cation intermixing between LaAlO_3 and SrTiO_3 is unavoidable at the interface¹⁹, it is likely that some additional electrons trapped near the interface region of LaAlO_3 are also included in the inline holography data.

Revised manuscript (page 10):

We note that the incremental charges detected by the change in the Ti valence state (Ti^{3+}) from EELS Ti-L_{2,3} differ from those measured by inline electron holography, as these two techniques probe fundamentally different types of charges. The inline electron holography measures the net charge density, encompassing all charges, whereas the charges detected by the change of Ti valence state to Ti^{3+} from EELS Ti-L_{2,3} edge specifically reflect electrons localized within the Ti-3d orbitals. Consequently, the incremental 2DEG density assessed by the change of Ti valence state from EELS Ti-L_{2,3} edge is typically smaller than that measured by inline electron holography. The latter captures a broader spectrum of net charges distributed across the somewhat blurred $\text{LaAlO}_3/\text{SrTiO}_3$ interface, including some additional charges trapped near the interface region of LaAlO_3 .

REVIEWERS' COMMENTS

Reviewer #2 (Remarks to the Author):

The Authors did change the interpretation of EELS data accepting my suggestions. With these changes I reccomend pubblication of the manuscript.

REVIEWER COMMENTS

Reviewer #2 (Remarks to the Author):

The Authors did change the interpretation of EELS data accepting my suggestions. With these changes I recommend publication of the manuscript.

Reply: We sincerely appreciate the reviewer's insightful comments and advice concerning the interpretation of EELS data and its underlying fundamental principles, which guided us toward an accurate assessment of the data.